# Changes in Arctic Ocean plankton community structure and trophic dynamics on seasonal to interannual timescales

Gabriela Negrete-García[1], Jessica Y. Luo[2], Colleen M. Petrik[2], Manfredi Manizza[1], and Andrew D. Barton[1,3]

[1]Scripps Institution of Oceanography, University of California San Diego, La Jolla, California, United States of America
[2]Geophysical Fluid Dynamics Laboratory, National Oceanic and Atmospheric Administration, Princeton, New Jersey, United States of America
[3]Department of Ecology, Behavior and Evolution, University of California, San Diego, California, United States of America

**Correspondence:** g1negret@ucsd.edu

**Abstract.** The Arctic Ocean experiences significant seasonal to interannual environmental changes, including in temperature, light, sea ice, and surface nutrient concentrations, that influence the dynamics of marine plankton populations. Here, we use a hindcast simulation (1948-2009) of size-structured Arctic Ocean plankton communities, ocean circulation, and biogeochemical cycles in order to better understand how seasonal to interannual changes in the environment influence phytoplankton physiology, plankton community structure, trophic dynamics, and fish production in the Arctic Ocean. The growth of model phytoplankton was primarily limited in winter, spring, and fall by light, but in summer, the growth of smaller and larger phytoplankton was mostly limited by temperature and nutrient availability, respectively. The dominant trophic pathway in summer was from phytoplankton to herbivorous zooplankton, such that the average trophic position of model zooplankton was lower in the summer growing season compared with the rest of the year. On interannual timescales, changes in plankton community composition were strongly tied to interannual changes in bottom-up forcing by the environment. In the summer, in years with lower ice and warmer temperatures, the biomass of phytoplankton and zooplankton was higher, the size abundance relationship slopes were more negative (indicative of a phytoplankton community enriched in smaller phytoplankton), zooplankton had higher mean trophic position (indicative of greater carnivory), and potential fisheries production was greater, fueled by increased mesozooplankton biomass and flux of organic matter to the benthos. The summertime shift toward greater carnivory in warmer and low-ice years was due primarily to changes in phenology, with phytoplankton and microzoopankton blooms occurring approximately one month earlier in these conditions, and carnivorous zooplankton increasing in abundance during summer. The model provides a spatially and temporally complete overview of changes in plankton communities in the Arctic Ocean occurring on seasonal to interannual timescales, and provides insights on the mechanisms underlying these changes as well as their broader biogeochemical and ecosystems significance.

## 1 Introduction

The Arctic Ocean is a large and heterogeneous ocean basin, consisting of multiple regions with distinct environmental conditions, plankton community structure, and biogeochemical processes (Carmack and Wassmann, 2006; Michel et al., 2015).

Seasonal variations due to cold, dark winters and relatively warm, sunlit summers lead to shifts in light, temperature, nutrient, and sea ice conditions throughout the year (Wassmann et al., 2011; Wassmann and Reigstad, 2011). These seasonal changes regulate the growth of phytoplankton and determine the initiation and extent of phytoplankton blooms throughout the Arctic (Sakshaug, 2004; Sherr et al., 2003). Light entering the surface ocean changes with solar elevation, sea ice thickness, sea ice snow cover, and other factors (Nicolaus et al., 2012; Castellani et al., 2022). The increase in the transmission of sunlight into the upper ocean in late spring and summer, coupled with the cold-season accumulation of nutrients from nutrient-rich deep waters into the surface layer (Tremblay and Gagnon, 2009), create favorable conditions for phytoplankton to bloom. Phytoplankton growth may become nutrient limited as nutrients are depleted in summer, and decreasing light and temperature during fall and winter can lead to light and temperature limitation, which persists until the following growing season (Danielson et al., 2017; Lowry et al., 2015).

In addition, the Arctic Ocean experiences interannual changes in the environment which have profound impacts on marine primary productivity (Arrigo et al., 2008; Arrigo and van Dijken, 2011; Ardyna et al., 2014; Fujiwara et al., 2018). While natural modes of climate variability are evident in the Arctic Ocean (e.g., Limoges et al., 2020), in recent decades the magnitude and impacts of anthropogenic climate change have been increasing (Ardyna and Arrigo, 2020). The shift from thick multiyear ice to thinner ice in the past decades has extended the growing season and expanded the open water suitable for phytoplankton growth, resulting in an increase in Arctic primary production due to greater light availability and vertical mixing (Arrigo et al., 2008; Arrigo and van Dijken, 2011; Overland et al., 2014; Serreze et al., 2007; Overland and Wang, 2013; Comiso, 2012). As temperatures in the Arctic warm, limitation on growth imposed by cold temperatures weakens, likely increasing primary production (Eppley, 1972; Bopp et al., 2013). Meanwhile, increased water-column stratification from warmer temperatures, increased precipitation, runoff from land and rivers, and ice melt might alter surface water stability (Bethke et al., 2006; Carmack and McLaughlin, 2011), confining cells to the surface layer and increasing exposure to light. The stability of the surface layer also modulates nutrient supply to the euphotic zone resulting in reduced nutrient fluxes and primary productivity (Carmack et al., 2004; Tremblay and Gagnon, 2009). Additionally, changes in the biomass of phytoplankton species, such as diatoms, may impact both the quantity and quality of secondary production, potentially resulting in disruptions to the food web (Leu et al., 2011; Kohlbach et al., 2016; Duerksen et al., 2014; Schmidt et al., 2018).

To date, however, we have incomplete understanding of how seasonal to interannual changes in the dynamic and spatially-heterogeneous Arctic Ocean environment influence phytoplankton physiology, plankton community structure, trophic dynamics, and fish production. Spatially and temporally complete observations of Arctic Ocean plankton communities and ecosystems are still developing. This study addresses this gap by developing a whole-Arctic simulation of plankton communities, biogeochemical cycles, ocean circulation, and sea ice using state-of-the-art modeling tools. We use a hindcast (1948-2009) physical simulation from the MARBL-SPECTRA model simulating 62 years of ocean variability using historical forcing from the CORE-II reanalysis (Griffies et al., 2009; Large and Yeager, 2009). Layered onto this simulation, MARBL-SPECTRA (Negrete-García et al., 2022) simulates a diverse range of phytoplankton and zooplankton and their biotic interactions, as well as interactions with biogeochemical cycles. The coupled physical, ecological, and biogeochemical models allow for the mechanistic representation of important marine plankton community properties such as phenology, biogeography, and trophic

transfers, as well as the coupled cycles of carbon, nitrogen, phosphorous, iron, silicon and oxygen (Negrete-García et al., 2022), all embedded in a three dimensional ocean circulation model. With MARBL-SPECTRA, we can investigate how plankton community structure and trophic food webs change in the Arctic on seasonal to interannual timescales, the mechanisms shaping these dynamics, and how these changes affect biogeochemical cycles and fish production.

## 2 Methods

### 2.1 MARBL-SPECTRA model

In this study, we used simulations with a global ocean biogeochemical and plankton community model called MARBL-SPECTRA (Negrete-García et al., 2022), an extension of the Marine Biogeochemistry Library (MARBL) model (Long et al., 2021), coupled to the ocean-ice models POP2-CICE5 (Smith et al., 2010; Hunke et al., 2017). MARBL is an intermediate-complexity ocean biogeochemical model that resolves the elemental cycles of C, N, P, O, Fe, and Si, and serves as the ocean biogeochemical component model of the Community Earth System Model v2 (Danabasoglu et al., 2020). In an ocean-ice-biogeochemistry configuration, POP2-CICE5-MARBL-SPECTRA simulates ocean circulation, sea ice, biogeochemical cycles, and marine plankton communities, as well as their interactions. MARBL-SPECTRA extends the simple plankton food web in MARBL to include substantially more plankton functional types and size classes, providing a more complex food web structure capable of examining temporal and spatial patterns in plankton ecosystem size structure, food web length and phenology, and how these influence ocean biogeochemical cycles. Additionally, the explicit representation of microzooplankton and mesozooplankton size classes allows for a better resolution of top-down controls on plankton community structure and productivity (Negrete-García et al., 2022).

The plankton community within MARBL-SPECTRA includes nine phytoplankton groups belonging to four different plankton functional types (picoplankton, mixed phytoplankton, diatoms, and diazotrophs) and six zooplankton groups divided into two microzooplankton ($<200$ $\mu$m ESD) and four mesozooplankton size classes (between 0.2 mm and 20 mm) (Negrete-García et al., 2022, Figure 1). In this context, mixed phytoplankton includes implicit calcifiers and solitary protists not included in the other functional groups, such as picoeukaryotes and autotrophic dinoflagellates. The traits and interactions of different sizes of phytoplankton are determined by size-based contrasts on nutrient uptake, light harvesting, carbon-to-chlorophyll ratios, and susceptibility to zooplankton grazing (Negrete-García et al., 2022). For instance, phytoplankton maximum specific growth rates decrease as their body size increase, while variations in maximum specific growth rates persist among different phytoplankton functional groups (Edwards et al., 2012). Because predator-to-prey feeding relationships are tied to the structure and energy flows within a plankton food web, MARBL-SPECTRA implements a feeding kernel similar to the Laplace distribution with predator-to-prey size ratios increasing and the feeding kernel broadening as predator size increases (Hansen et al., 1994).

MARBL-SPECTRA allows for the mechanistic representation of important marine plankton community properties such as phenology, biogeography, and trophic transfer (Negrete-García et al., 2022). For example, model picoplankton and small mixed phytoplankton are more successful in the low nutrient subtropical gyres, whereas large diatoms and mixed phytoplankton are more competitive in the highly seasonal, high nutrient oceans (Negrete-García et al., 2022). MARBL-SPECTRA can simulate

phenology and succession in a more diverse fashion than other plankton community models with fewer taxa(Moore et al., 2004; Long et al., 2021). In addition, the model simulates how predator-prey dynamics and trophic efficiency vary across gradients in total ecosystem productivity. Shorter food chains that export proportionally more carbon from the surface to the ocean interior occur in productive, eutrophic regions, whereas in oligotrophic regions, the food chains are relatively long and export less organic matter from the surface (Negrete-García et al., 2022). The ocean-ice models (POP2-CICE5) coupled to MARBL-SPECTRA do not simulate algal growth and movement within sea ice, which is known to influence Arctic Ocean primary production (Sakshaug and Slagstad, 1991; Mundy et al., 2009; Arrigo et al., 2014). MARBL-SPECTRA balances ecological realism and computational cost, providing plausible simulations of key ecosystem and biogeochemical processes. This balance is crucial for studying the Arctic Ocean, where complex interactions between plankton communities and physical environmental factors play significant roles in ecosystem dynamics. The model's ability to represent diverse plankton functional types and their responses to varying nutrient and light conditions enable a nuanced understanding of Arctic biogeochemical cycles and food web structures. For a comprehensive description of MARBL-SPECTRA and a diagram showing the model structure, see Negrete-García et al. (2022).

### 2.1.1 Growth limitation terms

We examined the factors that limit model phytoplankton growth in order to understand how and why phytoplankton physiology changes in space and time in the Arctic Ocean. Nutrients, light, and temperature are essential for phytoplankton growth and play important roles in structuring phytoplankton productivity throughout the global ocean. The realized growth rate (eq. 1) of each model phytoplankton type ($i$) is function of the maximum photosynthetic rate at the reference temperature ($PC_i^{ref}$) of 293.15°K, multiplied by limitation terms for light ($\gamma_i^I$), temperature ($\gamma_i^T$), and nutrients ($\gamma_i^N$). We assessed which of these factors were most limiting by selecting the lowest biomass-weighted limitation value between each limitation term for each phytoplankton group in each grid cell throughout the Arctic Ocean.

$$growth = PC_i^{ref} \gamma_i^T \gamma_i^N \gamma_i^I P_i \tag{1}$$

Following Leibig's Law of the Minimum (Eq. 2), nutrient limitation on growth ($\gamma_i^N$) was determined based on the most limiting nutrient (e.g., N, P, Si, or Fe) for each phytoplankton. The effect of growth rate of each nutrient required by each phytoplankton was represented according to Michaelis-Menten kinetics (eq. 3), where $\kappa_i$ represents the half-saturation nutrient concentration for each phytoplankton type according to allometric relationships defined by Edwards et al. (2012) and described in more detail in Negrete-García et al. (2022).

$$\gamma_i^N = min(N_{N,i}^{lim}, N_{P,i}^{lim}, N_{Fe,i}^{lim}, N_{Si,i}^{lim}) \tag{2}$$

$$N_i^{lim} = \frac{N}{N + \kappa_i} \tag{3}$$

From the phytoplankton-specific limiting nutrient, we calculated the biomass-weighted mean of the most limiting nutrient resource for each phytoplankton size class within each phytoplankton group (e.g., diatoms, mixed phytoplankton). Picoplankton, mixed phytoplankton, and diazotrophs do not require silicate, and diazotrophs do not require nitrate or ammonium.

The light sensitivity of growth rate ($\gamma_i^I$; Eq. 4) assumes that photosynthesis per unit of biomass saturates with increasing irradiance. This calculation takes into account parameters such as $PC_i^m$ (carbon-specific, light-saturated photosynthesis rate), $\alpha_i^{Chl}$ (Chl-specific initial slope of the photosynthesis-irradiance curve), $I$ (instantaneous irradiance), and $\theta_i^C$ ($Chl$:C ratio). The light limitation was determined using the phytoplankton biomass-weighted mean of the upper-ocean light limitation term for each phytoplankton type. Considerable spatial heterogeneity exists in sea ice thickness, which affects light available for phytoplankton growth below sea ice. Following the approach of Long et al. (2015), our model calculated phytoplankton growth limitation terms for the distribution of sea ice thicknesses present within the model grid cell, and then averaged these values over the grid cell to estimate the average light limitation in the grid cell.

$$\gamma_i^I = 1 - e^{\frac{-\alpha_i^{Chl}\theta_i^C I}{PC_i^m}} \tag{4}$$

For temperature modulation of growth rate ($\gamma_i^T$; Eq. 5), MARBL-SPECTRA employs an Arrhenius-Van't Hoff equation (Arrhenius, 1915), which expresses temperature dependence via activation energy ($E_a$) (Negrete-García et al., 2022). This equation accounts for picoplankton's higher temperature sensitivity compared to larger phytoplankton (Chen et al., 2014; Stawiarski et al., 2016), with $k$ representing Boltzmann's constant ($k = 8.617 \times 10^{-5}$ eV K$^{-1}$), $T$ being temperature, and $T_0$ denoting the model's reference temperature (293.15°K) (Negrete-García et al., 2022). Similarly to light, temperature limitation was calculated using the phytoplankton biomass-weighted mean of the upper-ocean temperature limitation term for each phytoplankton type. For further details on each parameter and the allometric relationships used to describe each phytoplankton type, see Negrete-García et al. (2022).

$$\gamma_i^T = e^{(\frac{-E_a(T_0-T)}{kT_0T})} \tag{5}$$

### 2.1.2 Predation fluxes

We estimated the predation fluxes, defined as the rate of carbon consumed for each plankton per unit volume of water (mmol C m$^{-3}$ d$^{-1}$), to define the structure of the model plankton food web and examine how and why this structure changes through space and time in the Arctic Ocean. The biomass of phytoplankton in the MARBL-SPECTRA model is influenced by a balance between growth and losses due to grazing, mortality, and aggregation (Negrete-García et al., 2022). In the model, predation on phytoplankton (mmol C $m^{-3}$ $d^{-1}$) is represented by a Holling type II functional response (Negrete-García et al., 2022), which describes how the rate of predation pressure increases approximately linearly as prey density increases, before reaching a maximum rate at high prey concentrations. To calculate predation fluxes, we summed all the rates of phytoplankton grazing by micro- and mesozooplankton for each phytoplankton or zooplankton size class, and compared their magnitudes.

## 2.2 Model simulations

In order to analyze how and why plankton communities in the Arctic Ocean change on seasonal to interannual timescales, we examined environmental, biogeochemical, and plankton community model output from the last 62 years of a global simulation of the MARBL-SPECTRA model (Negrete-García et al., 2022). MARBL-SPECTRA was run within a coupled ocean-ice simulation using the Parallel Ocean Program, version 2 (POP2) model (Smith et al., 2010), and the sea-ice model CICE5 (Hunke et al., 2017), forced with two cycles of the 62-year Common Ocean-Ice Reference Experiment (CORE-II) data set (Large and Yeager, 2009). This differs from the phase 2 of the Ocean Model Intercomparison Project (OMIP-2) protocol, where the forcing undergoes five repeating cycles (Griffies et al., 2016). A shorter integration does not provide a fully-equilibrated model solution in the deep ocean, but has been used for studying surface ocean dynamics (Stock et al., 2014). Thus, by the end of the second cycle of the 62 year spin up time, surface biomass distributions are nearing an equilibrium state, even if deep ocean tracers may not be. The seasonal analysis was conducted using the final 20 years (1990 - 2009) of the second 62-year cycle, while the interannual analysis was done using the full 62 years.

These CORE-II data sets, spanning from 1948 to 2009 with monthly resolution, provide global information on air-sea fluxes of momentum, heat, and freshwater (Large and Yeager, 2009), and have served as a standardized set of common atmospheric boundary conditions, widely adopted by the ocean modeling community. They have been employed to force and compare various coupled ocean-sea ice models used in the Coupled Model Intercomparison Project Phase 5 (CMIP5) experiments (Ilıcak et al., 2016; Wang et al., 2016a, b). The MARBL-SPECTRA simulation therefore is a hindcast simulation that represents seasonal to interannual changes in the environment and plankton community. Though the CORE-II forcing captures the historical climate signal, we instead focus on seasonal-to-interannual variability because it is the larger signal compared to climate change during this time period. The model has a horizontal resolution of $1^o$ with 60 vertical depth levels ranging in thickness from 10 m in the upper 150 m to 250 m in the deep ocean (Moore et al., 2013). Riverine nutrient fluxes (N, P, Si, Fe), dissolved inorganic carbon, alkalinity, and DOM fluxes were integrated into the CESM2 ocean model using estimates from GlobalNEWS (Mayorga et al., 2010) and the Integrated Model to Assess the Global Environment-Global Nutrient Model (IMAGE-GNM) (Beusen et al., 2015, 2016). Nutrient inputs from rivers encompass dissolved inorganic nitrogen (DIN), phosphorus (DIP), Si, and Fe, along with dissolved organic nitrogen and phosphorus, while carbon inputs included both inorganic and organic forms. MARBL-SPECTRA qualitatively reproduces observed global patterns of surface nutrients, chlorophyll biomass, net primary production, and the biogeographies of a range of plankton size classes, as well as how predator-prey dynamics and trophic efficiency vary across gradients in total ecosystem productivity (Negrete-García et al., 2022).

To characterize regional differences in plankton community structure, trophic dynamics, and potential fish production, we divided the Arctic Ocean into ten geographic sectors where appropriate. The central Arctic is defined as the region stretching from the North Pole to 80°N. The rest of the geographic sectors stretching from 80°N to 60°N were demarcated by longitude (Figure 1) and include the Chukchi Sea (180$^o$W to 155$^o$W), Beaufort Sea (155$^o$W to 125$^o$W), Canadian Archipelago (125$^o$W to 70$^o$W), Baffin Sea (70$^o$W to 45$^o$W), Nordic Seas (45$^o$W to 20$^o$E), Barents Sea (20$^o$E to 53$^o$E), Kara Sea (53$^o$E to 90$^o$E), Laptev Sea (90$^o$E to 145$^o$E) and East Siberian Sea (145$^o$E to 180$^o$W) sectors (Fig. 1). When examining seasons, months were

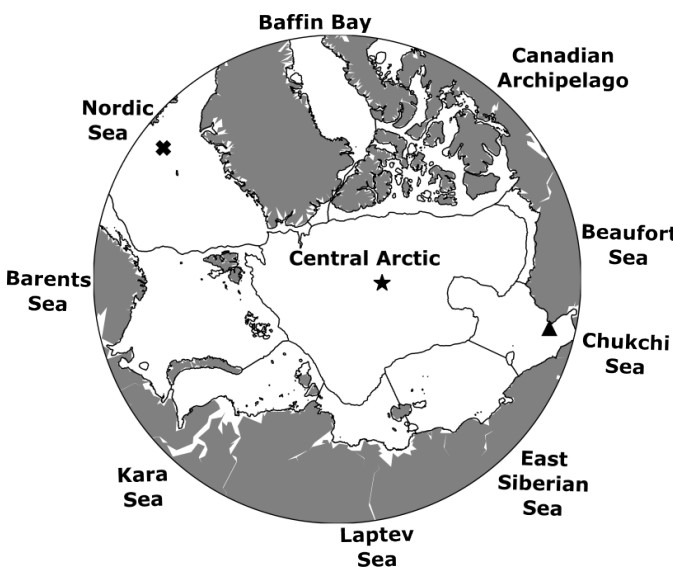

**Figure 1.** Map of the Arctic Ocean divided into ten sections included in the analysis, following the Lewis et al. (2020) regional mask. Additionally, symbols represent three grid cells selected throughout the Arctic Ocean for growth limitation analysis. The "x" symbol shows the Western Nordic Seas ($68.5^oN$, $348^oE$), the star symbol shows the central Arctic Ocean location ($85.5^oN$, $200^oE$), and the triangle symbol shows the Chukchi Sea location ($68.5^oN$, $168^oW$).

grouped as follows: December–February for boreal winter; March-May for boreal spring; June–August for boreal summer; September–November for boreal fall.

### 2.3 Size abundance relationship

We examined the model size abundance relationship as a means of understanding how and why the composition of the plankton community changes through space and time in the Arctic Ocean. Organism size is a crucial factor that shapes marine food webs by influencing the trophic organization of plankton communities and their biogeochemical functions (Andersen et al., 2016; Serra-Pompei et al., 2022). Smaller organisms are generally more abundant than larger ones, such that the abundance ($N$) of an organism of a particular size ($V$) is described by ($N = \alpha V^{\beta}$), with varying intercept ($\alpha$) and a slope ($\beta$) that is generally negative (Cermeño et al., 2006). For each location and every month (1990-2009 average), we calculated the slope and intercept of the size-abundance relationship by performing a linear least-squares regression between the logarithmic abundances of all phytoplankton size classes and their corresponding biovolumes (in $\mu m^3$). To estimate phytoplankton abundance (in cells $m^{-3}$), we divided the vertically-integrated phytoplankton biomass ($\mu g$ C $m^{-3}$) over the top 150 meters of the water column by the mass of each phytoplankton cell ($\mu g$ C cell$^{-1}$).

The slope ($\beta$) and intercept ($\alpha$) of the size abundance relationship together describe the community size structure. Locations with a more negative size abundance slope ($\beta$) indicate that the phytoplankton community is more dominated by smaller phytoplankton cells; this can occur either by having very few large phytoplankton or proportionally more small phytoplankton.

Alternatively, locations with a less negative slopes indicate either a higher contribution of larger phytoplankton, or a lower contribution of smaller phytoplankton. More negative slopes are typically found in oligotrophic regions while less negative slopes are found in eutrophic regions (e.g., Huete-Ortega et al., 2010). Within MARBL-SPECTRA, regions of more (less) negative slopes tend to be less efficient at exporting carbon from the ocean surface, due to the weaker (greater) contribution of larger phytoplankton to sinking particle flux (Negrete-García et al., 2022). The intercepts are indicators of overall phytoplankton abundance, where lower intercept ($\alpha$) values often reflect low total phytoplankton abundance, while larger intercept values may represent greater phytoplankton abundance.

### 2.3.1 Interannual analysis

In order to assess how extremes in temperature, ice fraction, and $NO_3$ concentrations influence key ecosystem properties, we developed a criteria for identifying extremes in environmental forcing. These extremes were identified by years where the anomalies in temperature, ice fraction, and $NO_3$ concentration fell above or below the $90^{th}$ and $10^{th}$ percentile, respectively. The monthly anomalies for each variable were calculated by subtracting the long-term monthly climatology, which represents the average conditions for each month across the second 62-year CORE-II forced simulation period. The monthly anomalies were then aggregated into annual and seasonal averages, representing winter (December-February), spring (March-May), summer (June-August), and fall (September-November). Annual anomalies in temperature, ice fraction, and $NO_3$ concentration were employed to identify extreme anomaly years for fisheries production analysis, while seasonal anomalies were utilized for evaluating size abundance, trophic level, and plankton biomass. We then contrast the ecological metrics (e.g., total phytoplankton biomass) between the extreme positive ($>90^{th}$ percentile) or negative ($<10^{th}$ percentile) anomaly years.

### 2.4 Fisheries production

We estimated potential fisheries production in order to understand how and why it varies in response to changing environmental conditions and food web structure in the modeled Arctic Ocean. We used an empirical model developed by Stock et al. (2017) that relates fish production (gC m$^{-2}$ y$^{-1}$) to the production of mesozooplankton not consumed by other zooplankton ($MESOZP$; gC m$^{-2}$ y$^{-1}$) and the flux of organic matter to the benthos ($FDET$; gC m$^{-2}$ y$^{-1}$) :

$$FP = FDET * TE^{(TL_{eq} - TL_{FDET})} + MESOZP * TE^{(TL_{eq} - TL_{MESOZP})} \tag{6}$$

The trophic efficiency $TE$ varies from 0-0.4 across aquatic ecosystems (Pauly and Christensen, 1995), but here was set to a constant value of 0.2. $TL_{eq}$ is the average trophic level of the fish of interest. We mainly focus on fisheries of trophic level four, such as Arctic Cod (*Boreogadus saida*), that are often found in close association with sea ice and represent an important trophic link in the Arctic food webs (Jennings and Van Der Molen, 2015; Graham et al., 2014). The exponents are adjusted by the average trophic level of the food source, $TL_{FDET}$ and $TL_{MESOZP}$. $TL_{FDET}$ is always equal to 1 whereas $TL_{MESOZP}$ is the mean mesozooplankton trophic level was computed based on model output, taking into account an ingestion-weighted biomass. In this calculation, zooplankton exclusively grazing on phytoplankton were assigned to trophic level two, while the

assignment of higher trophic levels was determined by calculating the proportion of food obtained from each subsequent trophic level using ingestion-weighted biomass. This empirical estimate of fish production assumes that food for fish comes from multiple sources, where the structure of the exponents indicate the number of trophic transfers from the food source to the fishes in question.

## 2.5 Comparison between model and observations

We compared simulated surface temperature, salinity, nitrate, chlorophyll and mesozooplankton biomass distributions with available observations throughout the Arctic Ocean. We compared annual average surface (top 10 m) temperature, salinity, and nitrate in the model from 1990 to 2009 with surface observations from the World Ocean Atlas 2018 (https://www.ncei.noaa.gov/products/world-ocean-atlas; Locarnini et al., 2018; Zweng et al., 2019) from 1995 to 2004. Model annual mean surface (top 10m) chlorophyll (Chl) concentrations were compared with satellite-based chlorophyll estimates using an ocean color algorithm tailored for the Arctic Ocean developed by Lewis et al. (2020) from 1998-2007, which corresponds to the last nine years of the CORE-II forcing dataset and MARBL-SPECTRA integration. Modeled mesozooplankton summer average biomass over the top 150 m was compared with available summer observations from the NOAA COPEPOD global zooplankton database over the top 200 m, of which the global mesozooplankton carbon biomass dataset (Moriarty and O'Brien, 2013) is the most relevant and accessible for model output comparison. The paucity of mesozooplankton data in the Arctic Ocean makes model-observation comparisons challenging.

## 3 Results and Discussion

### 3.1 Comparison between model and observations

Overall, the model captured spatial gradients in surface temperature, salinity and nitrate in this region. The differences in temperature between the model and observations (maximum of +/- $1^oC$) were relatively small compared with spatial and temporal temperature gradients (range $\approx 10^oC$), and differences in salinity between the model and observations were highest in coastal regions with high influx of freshwater. MARBL-SPECTRA annual average surface temperature in the Barents and Nordic Seas was slightly colder compared with observations, while in the Central Arctic, the model was slightly warmer than observations (Fig. 2c). Model annual average surface salinity in the interior shelf regions of the Kara, Barents, East Siberian, and Beaufort Seas was greater than observed (Fig. 2f). Surface nitrate concentrations differed the most along the central Arctic and throughout the Barents and Baffin Bay (Fig. 2i). Specifically, the model illustrated a trend towards a more oligotrophic western Arctic Ocean basin (Fig. 8e). This phenomenon can be mainly attributed to the compounding impacts of sea-ice loss and increasing water column stratification, resulting in a diminished influx of nutrients from subsurface waters, a trend aligning with findings from observations Zhuang et al. (2021).

MARBL-SPECTRA generally underestimated surface chlorophyll along coastal waters above the Russian continental shelves compared to satellite-based chlorophyll estimates using an ocean color algorithm tailored for the Arctic Ocean developed by

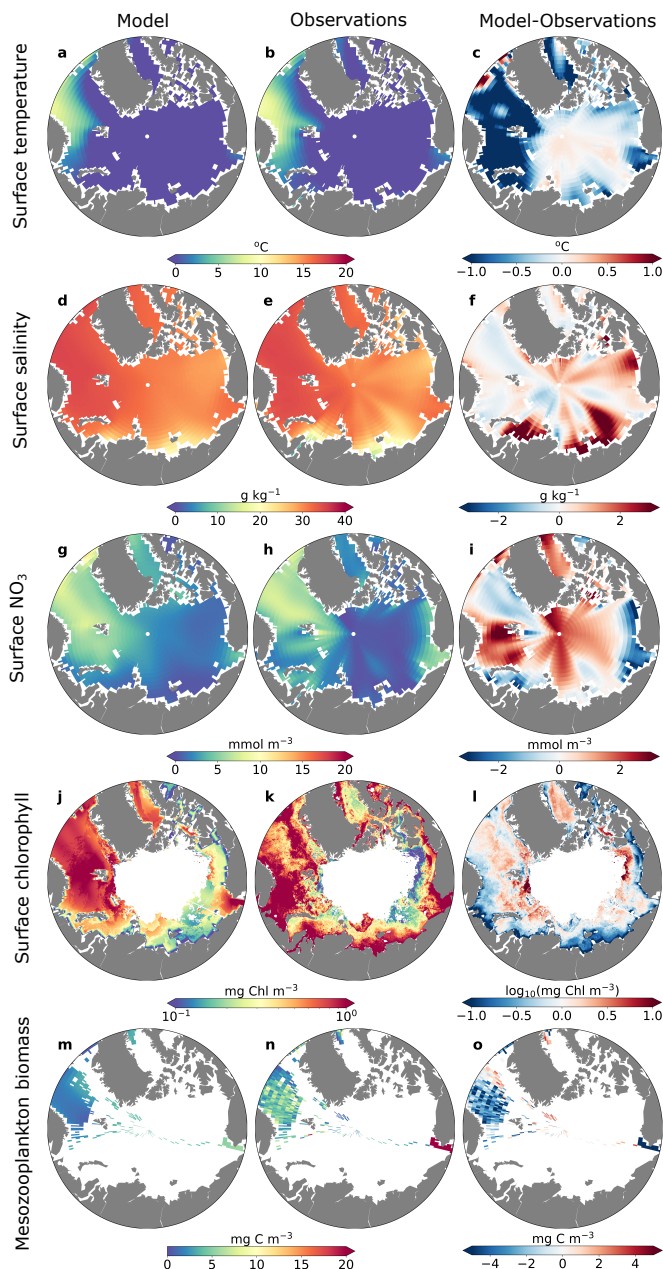

**Figure 2.** Annual average model (first column) and observed or satellite-estimated (second column) surface (top 10 m) temperature (a-b), surface (top 10 m) salinity (g kg$^{-1}$) (d-e), surface NO$_3$ (mmol $m^{-3}$) (g-h), surface chlorophyll ( log$_{10}$ mg Chl m$^{-3}$) (j-k), and summer mesozooplankton biomass (mg C m$^{-3}$) averaged over the top 150m (m-n). The third column shows differences between the model minus observations (c, f, i, o) and satellite-based estimates (l).

Lewis et al. (2020) (Fig. 2l). This underestimation may be attributed to inaccuracies of the satellite estimates from the atmospheric correction scheme, sensor calibration, or bio-optical algorithms, which were not optimized to account for the presence of colored dissolved organic matter (CDOM) in coastal waters (Siegel et al., 2002, 2013; Mustapha et al., 2012).

Though mesozooplankton data in the Arctic are somewhat limited, and comparing zooplankton observations with modeled zooplankton is not straightforward, MARBL-SPECTRA underestimated mesozooplankton biomass in most regions containing observations (Fig. 2o). Several potential factors could contribute to this underestimation, such as the allometrically-constrained zooplankton parameters lacking the necessary degrees of freedom to accurately simulate the full range of zooplankton dynamics (Negrete-García et al., 2022). Another possibility is that the feeding kernel used in the model may be overly broad (Negrete-García et al., 2022). Additionally, it is worth noting that MARBL-SPECTRA does not account for certain aspects of zooplankton life histories, including dormancy or diapause, which could be contributing to the observed discrepancies in mesozooplankton biomass estimates (Negrete-García et al., 2022).

Comparison between satellite and model chlorophyll is difficult due to known challenges of remote sensing in the Arctic Ocean, including but not limited to clouds, sea ice, and organic matter in the water column (Li et al., 2024; Gregg and Casey, 2007; Mikelsons and Wang, 2019). However, we further assessed the performance of MARBL-SPECTRA by comparing the seasonality of surface chlorophyll in different regions of the Arctic Ocean with satellite chlorophyll estimates tailored to the Arctic Ocean (Lewis et al., 2020) (Fig. 3). Additionally, we examined the seasonal chlorophyll profiles, comparing them with the modeled seasonal ice fraction and photosynthetically active radiation (PAR) over the surface layer (10 m) for each Arctic region. With the exception of the Chukchi (Fig 3a) and Barents Seas (Fig 3b), model and satellite chlorophyll magnitudes were qualitatively similar. The satellite and model seasonal phenology of chlorophyll were similar in some regions (e.g., the Nordic Sea (Fig. 3i)) but shifted in others (e.g., Baffin Bay (Fig. 3g), Barents Sea (Fig. 3b))) due to temporal discrepancies between model and remotely sensed Arctic Ocean in timing of sea ice retreat. MARBL-SPECTRA simulated a summer peak in chlorophyll during July in the Siberian (Fig. 3c), Laptev (Fig. 3d), Kara (Fig. 3e) Seas and Canadian Archipelago (Fig. 3h), coinciding with the highest average photosynthetically active radiation (PAR) over the surface layer and a rapid decrease in sea ice fraction. In the Barents Sea (Fig. 3b) and Baffin Bay (Fig. 3g), MARBL-SPECTRA simulated a peak in chlorophyll concentrations of lower magnitude than the satellite-estimate, and with a month delay. This delayed bloom in the model simulation could be due to later retreat in sea ice in the sea-ice model, causing light limitation in April and May where the satellite product estimates the Chlorophyll peaks to occur. Comparing the Central Arctic region with satellite-based estimates was challenging due to the limited chlorophyll information available, as this area remains mostly covered by ice throughout the year.

## 3.2 Seasonal changes in the Arctic Ocean

### 3.2.1 Plankton biomass & nutrient concentrations

Seasonal variations in phytoplankton, zooplankton, nutrient concentrations and flux of organic matter to the benthos across the Arctic Ocean (Fig. 4) revealed distinct patterns of ecosystem dynamics. Throughout the year, the Arctic Ocean experienced

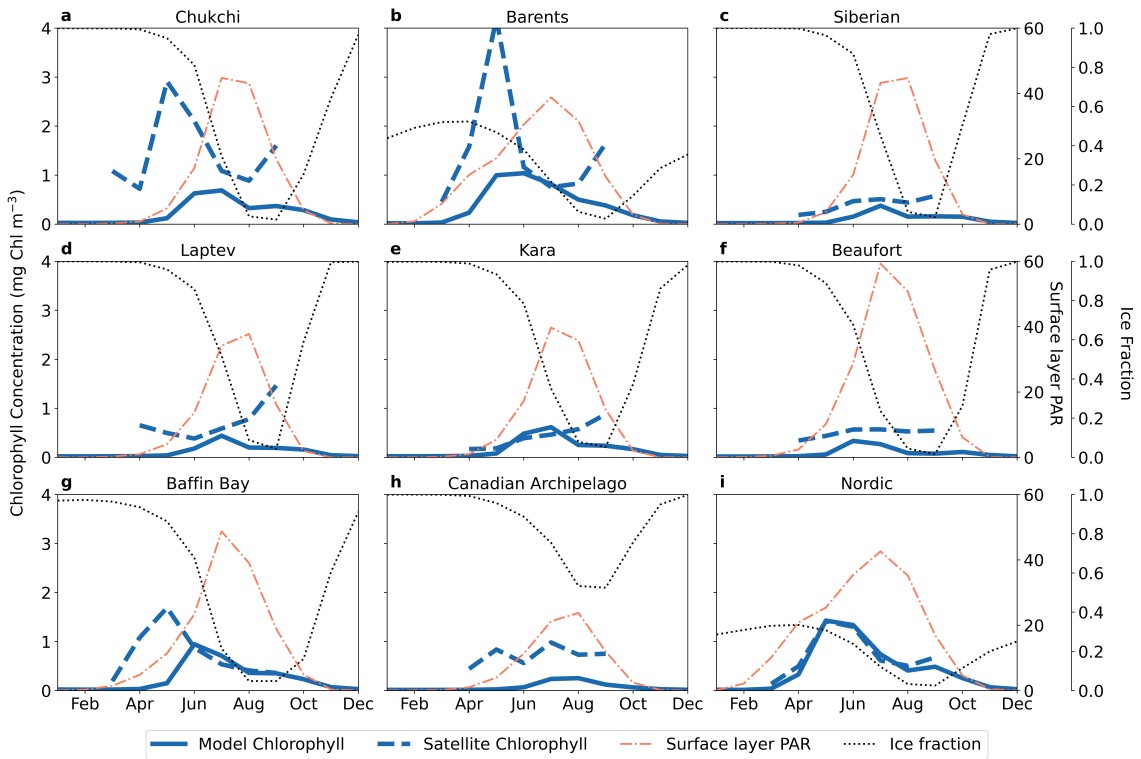

**Figure 3.** Modeled and satellite estimates of seasonal variability in surface chlorophyll. The solid blue lines depict the modeled monthly-averaged chlorophyll at the surface layer (10 m), while the dashed blue lines represent the satellite-estimated surface chlorophyll (Lewis et al. (2020)). Additionally, the dotted black line shows modeled monthly-averaged ice fraction, and the thin dashed red line represents the average photosynthetically active radiation (PAR) over the surface layer (10 m) (W m$^{-2}$). Seasonal cycles are displayed for nine different regions: the Chukchi Sea (a), Barents Sea (b), Siberian Sea (c), Laptev Sea (d), Kara Sea (e), Beaufort Sea (f), Baffin Bay (g), Canadian Archipelago (h), and Nordic Sea (i).

295  significant fluctuations in environmental conditions impacting the biological communities. Phytoplankton biomass was highest in summer (Fig. 4c) because of warmer temperatures and higher light, which occurred because of longer days and less sea ice. Phytoplankton blooms in the Nordic Seas started in spring instead of summer, corresponding to earlier onset of conditions favorable for phytoplankton growth. Zooplankton biomass was highest during summer (Fig. 4g). In contrast, winter and fall were characterized by reduced phytoplankton and zooplankton biomass (Fig. 4a,d, & e,h), consistent with diminished light

300  availability and lower temperatures, as well as nutrient limitation in certain regions in fall. Nutrient concentrations were high in the winter but low in the summer, reflecting accumulation of nutrients in the cold, dark winter and depletion in the summer (Fig. 4i). The flux of organic matter to the benthos ($FDET$) exhibited patterns mirroring those of phytoplankton biomass, with the highest fluxes occurring in summer (Fig. 4o) and the lowest in winter (Fig. 4m).

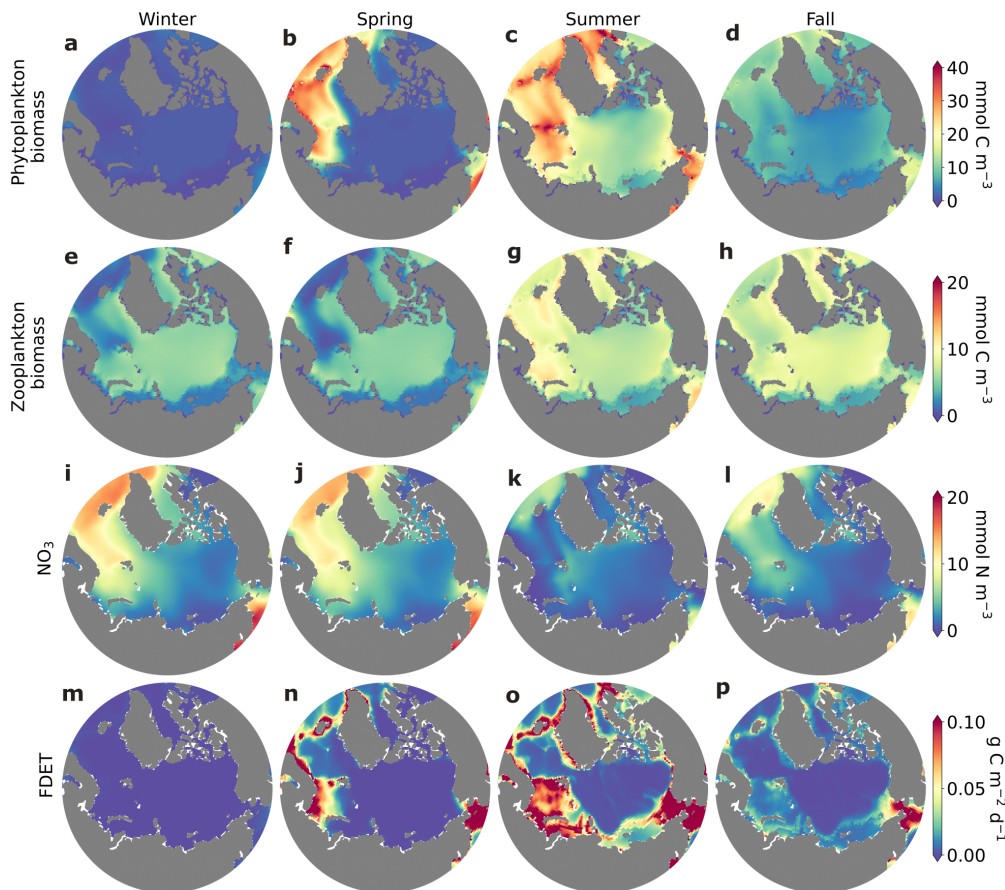

**Figure 4.** Surface (top 10m) phytoplankton (mmol C m$^{-3}$; a-d), zooplankton biomass (mmol C m$^{-3}$; e-h), NO$_3$ concentrations (mmol N m$^{-3}$; i-l), and flux of organic matter to the benthos ($FDET$; g C m$^{-2}$ d$^{-1}$; m-p) in winter (a,e,i; December-February), spring (b,f, j; March-May), summer (c,g, k; June-August), and fall (d,h,l; September-November). Data were averaged over the final 20 years of the model integration for each season.

### 3.2.2 Limitation of phytoplankton growth

305   Phytoplankton growth in the Arctic Ocean is influenced by nutrients, light, temperature, sea ice, sea ice snow cover, and other factors that vary seasonally (Lannuzel et al., 2020; Von Appen et al., 2021). In Figure 5, we assessed whether phytoplankton growth was most limited by light, temperature, or nutrients. In this case, nutrients can refer to limitation by nitrate, phosphate, silicate, or iron, but in practice in the model, phytoplankton growth was most often limited by nitrate (Supp. Fig. S2). During the winter months, low light conditions and higher sea ice cover caused light limitation of growth for all modeled phytoplankton 310   (Fig. 5a,e). In summer, increased freshwater from melting sea ice and runoff from land created a relatively stable, shallow mixed layer that confined cells to the surface and supported phytoplankton growth. Sufficient light and nutrients during the summer months enabled fast-growing phytoplankton to rapidly consume surface nutrients, leading to a decrease in nutrients

and restricting the growth of larger diatoms (Fig. 5g). Small phytoplankton experienced lower nutrient limitation (Fig. 5c) due to their greater capacity to acquire nutrients via diffusion relative to their nutrient demands (Edwards et al., 2012; Negrete-García et al., 2022). In fall, light levels decreased due to shorter days, sea ice accumulation, and snow accumulation on sea ice, and consequently phytoplankton growth became light-limited again (Fig. 5 d,h). During fall, if freeze-up is delayed and the sea surface is exposed to wind stress, increased wind-driven vertical mixing can be significant in promoting the growth of larger diatoms (Ardyna et al., 2014).

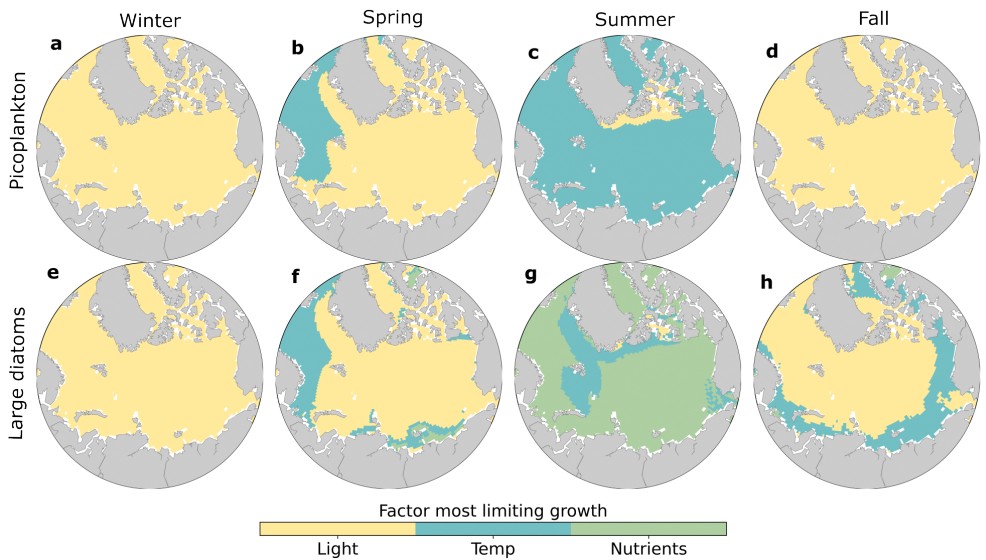

**Figure 5.** Phytoplankton growth limitation for picoplankton (a-d) and large diatoms (e-h) in winter (a,e; December-February), spring (b,f; March-May), summer (c,g; June-August), and fall (d,h; September-November). The growth limitation terms for light (yellow), temperature (blue), and nutrient (green) limitation were averaged over the three months in each season, averaged over 1990-2009 . Nutrient and light limitation terms were computed as biomass-weighted vertical averages of the top 100m. Temperature limitations were estimated using activation energy values for each phytoplankton type. Nitrate was the most limiting nutrient for phytoplankton growth for most regions and seasons (Supp. Fig. S2).

These model results showed that all model phytoplankton groups were most strongly limited by light in winter. In summer, the largest phytoplankton were most strongly limited by nutrients while the smallest phytoplankton were most strongly limited by temperature. These contrasting patterns of growth limitation in the model occurred because small phytoplankton have a higher competitive ability to acquire nutrients than do larger phytoplankton (e.g., Edwards et al., 2012). The strong and persistent limitation of Arctic Ocean phytoplankton growth in fall, winter, and spring allowed nutrients to accumulate in the surface (Fig. 4), promoting a strong summer bloom of phytoplankton and zooplankton, which is largely consistent with observations (Wassmann and Reigstad, 2011; Wassmann et al., 2011). Our results concerning factors limiting phytoplankton

growth are broadly consistent with previous modeling studies that have shown that temperature and light strongly limit phytoplankton growth in the Arctic Ocean (Krumhardt et al., 2020; Steinacher et al., 2010), however we provide additional context on seasonal changes and how limiting factors vary across phytoplankton sizes.

### 3.2.3 Biomass distribution and trophic fluxes in the plankton community

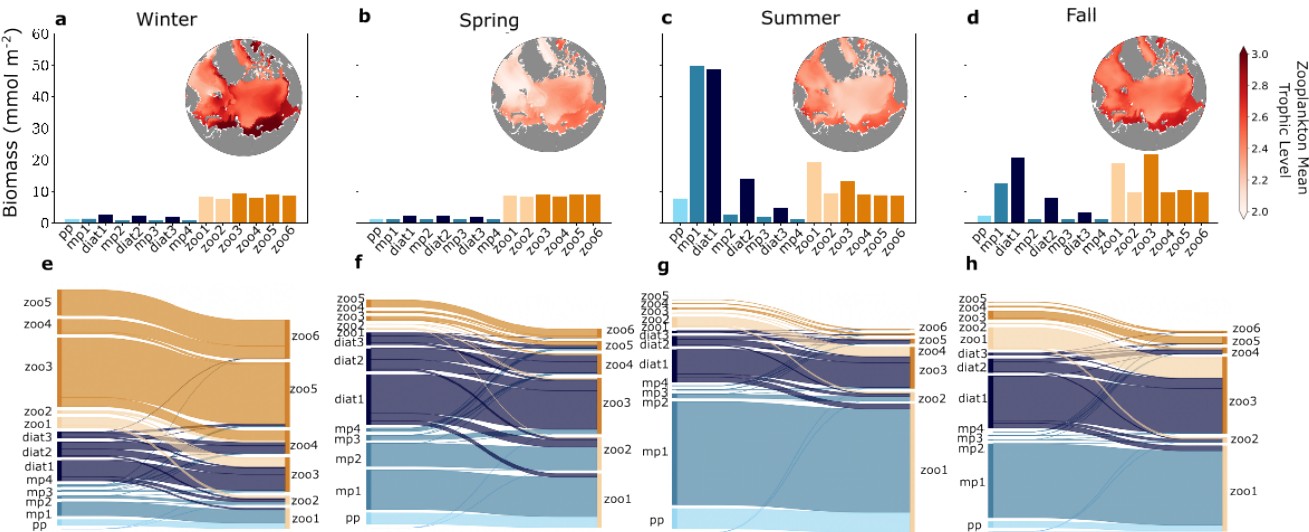

**Figure 6.** Plankton biomass (a-d) and grazing fluxes (e-h) between predators (right columns) and prey (left columns) averaged over winter (a,e), spring (b,f), summer (c,g), and fall (d,h) months across four phytoplankton functional groups - picoplankton (light blue), diazotrophs (sky blue), mixed phytoplankton (medium blue), and diatoms (dark blue), and two zooplankton functional groups - microzooplankton (light orange), and mesozooplankton (bright orange). Seasonal grazing fluxes (e-h) were calculated as proportions of the total for each season, reflecting the relative contribution of grazing within each specific season. Thicker lines represent a higher grazing flux from prey to predators, compared to thinner lines. The inset maps in (a-d) show the seasonal zooplankton mean trophic level averaged over the top 150 meters. The shading of the maps increases from light red to dark red, indicating a trophic level of 2 for entirely herbivorous zooplankton feeding on primary producers (light red) and a trophic level of 3 for carnivorous zooplankton that eat herbivores (dark red).

Seasonal variations in light, nutrient availability, and temperature strongly influenced the distribution of biomass in plankton communities and the dynamics of trophic interactions. During winter, total phytoplankton biomass was low (Fig. 4a-c), primarily due to light limitation of phytoplankton growth, and the community was mainly composed of diatoms and larger phytoplankton (Fig. 6a). Mesozooplankton populations survived by grazing on larger phytoplankton and other zooplankton (Fig. 6e), and the plankton food web was more carnivorous than in other seasons, as indicated by the relatively high average trophic position of mesozooplankton (Fig. 6a).

In spring, nutrients and light increased, promoting the growth of diatoms and mixed phytoplankton of small to medium size (Fig. 6b). This led to a modest increase in overall phytoplankton biomass, but caused a shift in trophic pathways, where a

greater proportion of the total trophic fluxes occurred between primary producers and zooplankton than in winter (Fig. 6f). Consequently, the average trophic position of zooplankton was lower than in winter (Fig. 6b).

In summer, the alleviation of light and temperature limitation on phytoplankton drove a rise in phytoplankton biomass across almost every size class, with the most significant increases occurring in the small to medium mixed phytoplankton (Fig. 6c). These small phytoplankton were the primary food source for zooplankton populations (Fig. 6g), and the dominant trophic pathways were from primary producers to small zooplankton. The average zooplankton trophic position was relatively low in spring and summer (Fig. 6c).

In fall, the decrease in sunlight angle and increased ice fraction led to increased light limitation and a decline in phytoplankton biomass (Fig. 6d). Despite this decline, small mixed phytoplankton and diatoms still made up a significant portion of biomass, and zooplankton populations were sustained by both phytoplankton and microzooplankton grazing, resulting in larger mean zooplankton trophic levels compared with summer (Fig. 6d).

The model indicated a pronounced seasonal shift from large to small phytoplankton driven by the seasonal reduction of
surface nitrate, which is largely consistent with observations from the region (Ardyna et al., 2017, 2011; Ardyna and Arrigo, 2020; Tremblay and Gagnon, 2009; Usov et al., 2024). The seasonal succession of zooplankton is significantly influenced by the size structure of the phytoplankton, consistent with Usov et al. (2024), which found that distinct seasonal groups of phytoplankton and zooplankton in the Chupa Inlet (White Sea) are interconnected, with smaller species playing a larger role in summer and autumn, enhancing trophic coupling throughout the seasonal cycle.

In addition, the model showed a large seasonal change in trophodynamics driven by bottom-up changes to the phytoplankton community. In summer, the predominant trophic transfer occurred between small to medium phytoplankton and their microzooplankton consumers (Fig. 6c,g), consistent with observations in the Arctic shelf region, where locations with higher chlorophyll fluorescence exhibited a greater prevalence of smaller zooplankton (Balazy et al., 2018). A large but regionally-variable fraction of summer phytoplankton production is consumed by microzooplankton in the Arctic Ocean and adjoining
seas (e.g., (Sherr et al., 2009, 2013; Yang et al., 2015)). In winter, model zooplankton biomass is greater than phytoplankton biomass, and the dominant trophic transfers are between zooplankton, such that the average zooplankton trophic level is relatively high (Fig. 6e). These seasonal shifts from net autotrophy in summer to net heterotrophy in other seasons are consistent with observations from the 2019–2020 MOSAiC International Arctic Drift Expedition in the Arctic Ocean (Chamberlain et al., 2024), and reflect seasonal variations in diet and survival strategies of more opportunistic zooplankton with adaptable feeding
behaviours (Berge et al., 2015), as observed by Choi et al. (2020) in *Oirhona similis* (copepod) exhibiting elevated trophic positions post-polar night. Model zooplankton do not diapause (see Section: Model study limitations), whereas many Arctic and high latitude copepods do (Baumgartner and Tarrant, 2017). Even with rates of metabolism reduced in winter (e.g., (Ikeda, 1985) and lower mass-specific metabolic rates associated with larger body size (Kiørboe and Hirst, 2014), larger zooplankton remaining in the winter water column would likely consume available zooplankton prey, or perhaps under-ice algae (Kohlbach
et al., 2016), to survive.

### 3.2.4 Phytoplankton community size structure

The size abundance relationship serves as an important metric for assessing the size distribution within plankton communities. Much like biomass distributions and trophodynamics, it is influenced by seasonal and regional changes in light, nutrients, and temperature. We found that less negative (or flatter) slopes generally occurred during the winter months (Fig. 7a), driven primarily by lower abundances of small phytoplankton in winter (lower intercept; Fig. 7e-h; Supp. Fig. S6). Conversely, steeper (more negative) slopes occurred during the summer months (Fig. 7c & Supp. Fig. S6), where less light limitation combined with sufficient nutrient concentrations encouraged the growth of small and medium phytoplankton (Fig. 5c,g).

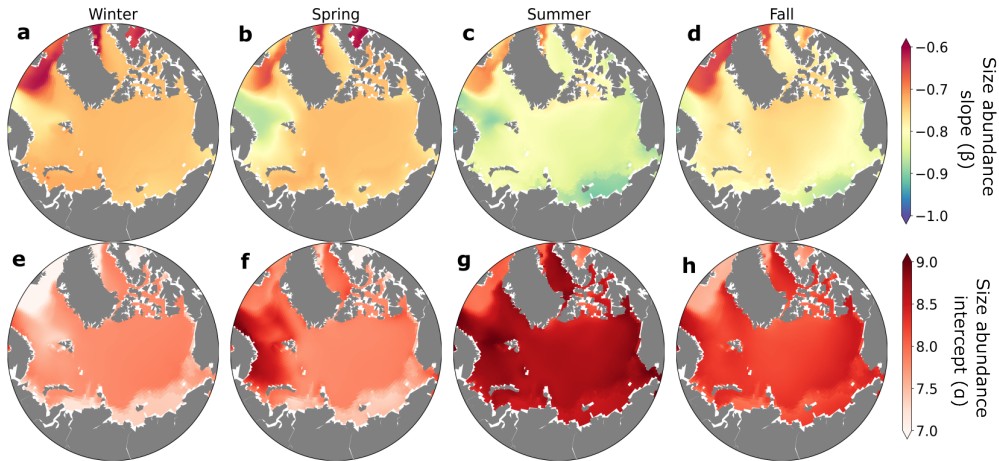

**Figure 7.** Seasonal dynamics of the size abundance slopes ($\beta$); a-d) and intercepts ($\alpha$; e-h) averaged over the top 150 meters for winter (a,e; December–February), spring (b,f; March-May), summer (c,g; June–August), and fall (d,h; September–November). The slopes and intercepts of the size abundance relationship were calculated using a linear least-squares regression between phytoplankton abundance in all size classes and their corresponding cell biovolume, on a log-log scale.

The increase of size abundance intercept in summer corresponded with well-documented increases in summer primary productivity and phytoplankton biomass in the region (Wassmann et al., 2011; Wassmann and Reigstad, 2011; Ardyna and Arrigo, 2020). The seasonal change in slope of the size abundance relationship in the modeled Arctic was driven primarily by seasonal variations in the abundance of smaller model phytoplankton, while the abundance of the larger groups remained relatively steady through time. Lampe et al. (2021) observed that, similar to the model results, seasonal changes in the plankton size spectra in the Fram Strait were strongly tied to changes in the smaller phytoplankton. However, observational and modeling studies focusing on lower latitude subtropical, subpolar, and coastal systems have often found that changes in the plankton size spectra were strongly tied to changes in the abundances of larger phytoplankton (San Martin et al., 2006; Ward et al., 2014; Marañón et al., 2018; Schartau et al., 2010). In these areas, larger phytoplankton were simulated to increase seasonally when nutrients were more abundant (Ward et al., 2012). While this mechanism also operates in the model, the effects on the size abundance relationship of seasonal alleviation of light limitation for smaller phytoplankton were typically larger. Small

phytoplankton in the model have a higher affinity for light compared with larger phytoplankton (Negrete-García et al., 2022),
which underpinned their highly dynamic responses to changes in light through time.

## 3.3 Interannual changes in the Arctic Ocean

### 3.3.1 Phytoplankton biomass and growth limitation

Phytoplankton biomass and the factors that limit growth vary interannually in response to environmental changes (Supp. Figs.
S3, S4). We focused our analysis on summer interannual changes in phytoplankton growth limitation, because it was the
season with the greatest differences in growth limitation among differing phytoplankton size classes (Fig. 5). When looking
at the Arctic Ocean as a whole, total summer model phytoplankton biomass was positively correlated with temperature (Fig.
8a) because the growth of picoplankton (pp) and small to medium-sized mixed phytoplankton (mp1, mp2), the groups which
contribute most to biomass in summer, was limited by temperature (Fig. 8b). Growth of the largest phytoplankton (diat3, mp4)
was limited by nutrients each summer, whereas for intermediate sizes (diat1, diat2, mp3), limitation switched between nutrients
and temperature interannually (Fig. 8b).

To better understand why phytoplankton communities responded differently in these regions, we selected three locations in
the Arctic Ocean for further discussion (Fig. 1): the Western Nordic Sea, Chukchi Sea, and Central Arctic Ocean.

The Western Nordic Seas experienced strong temperature, light, and nutrient variability (Fig. 8c). Growth of most phyto-
plankton in this region was limited by nutrients during summer months (Fig. 8d), except picoplankton (pp), whose growth
was primarily restricted by temperature (Fig. 8d). Small and medium-sized diatoms (diat1, diat2) were particularly affected by
interannual nutrient fluctuations, altering the strength of nutrient limitation (Fig. 8d).

The Chukchi Sea exhibited a positive correlation between temperature and biomass (Fig. 8e), primarily because growth
of small and medium-sized phytoplankton, which were major contributors to summer biomass, were limited by temperature.
Similar to the western Nordic Seas (Fig. 8d), the growth of most phytoplankton was limited by nutrients in summer, except
picoplankton (pp), whose growth was limited by temperature (Fig. 8f). However, small and medium-sized diatoms (diat1, diat2)
and mixed phytoplankton (mp1, mp2, mp3) were sensitive to interannual nutrient fluctuations, and switched from nutrient to
temperature limitation during years of elevated nutrient concentrations (e.g., 1976-1978).

The Central Arctic Ocean exhibited positive temperature, ice fraction, and nutrient anomalies toward the end of the model
simulation (Fig. 8g) resulting in greater variability in the growth limitation of most small and medium-sized phytoplankton.
Growth of the largest phytoplankton (diat3, mp4) was strongly limited by nutrients each summer, whereas the growth limitation
of the medium sized diatom (diat2) alternated between nutrients and temperature (Fig. 8h). The growth of the remaining
smaller-sized phytoplankton (pp, mp1, diat1, mp2, mp3) was primarily limited by temperature, with occasional shifts to nutrient
or light limitation. This was apparent in years with high ice fraction, (e.g., 1990 and 1997), where small and medium-sized
mixed phytoplankton (mp1, mp2) and small diatoms (diat1) switched their growth limitation from temperature to light.
The focus on three distinct Arctic Ocean regions—Western Nordic Sea, Chukchi Sea, and Central Arctic Ocean—revealed
dynamics of growth limitation influenced by temperature, light, and nutrient variations. In the Western Nordic Sea, nutrient

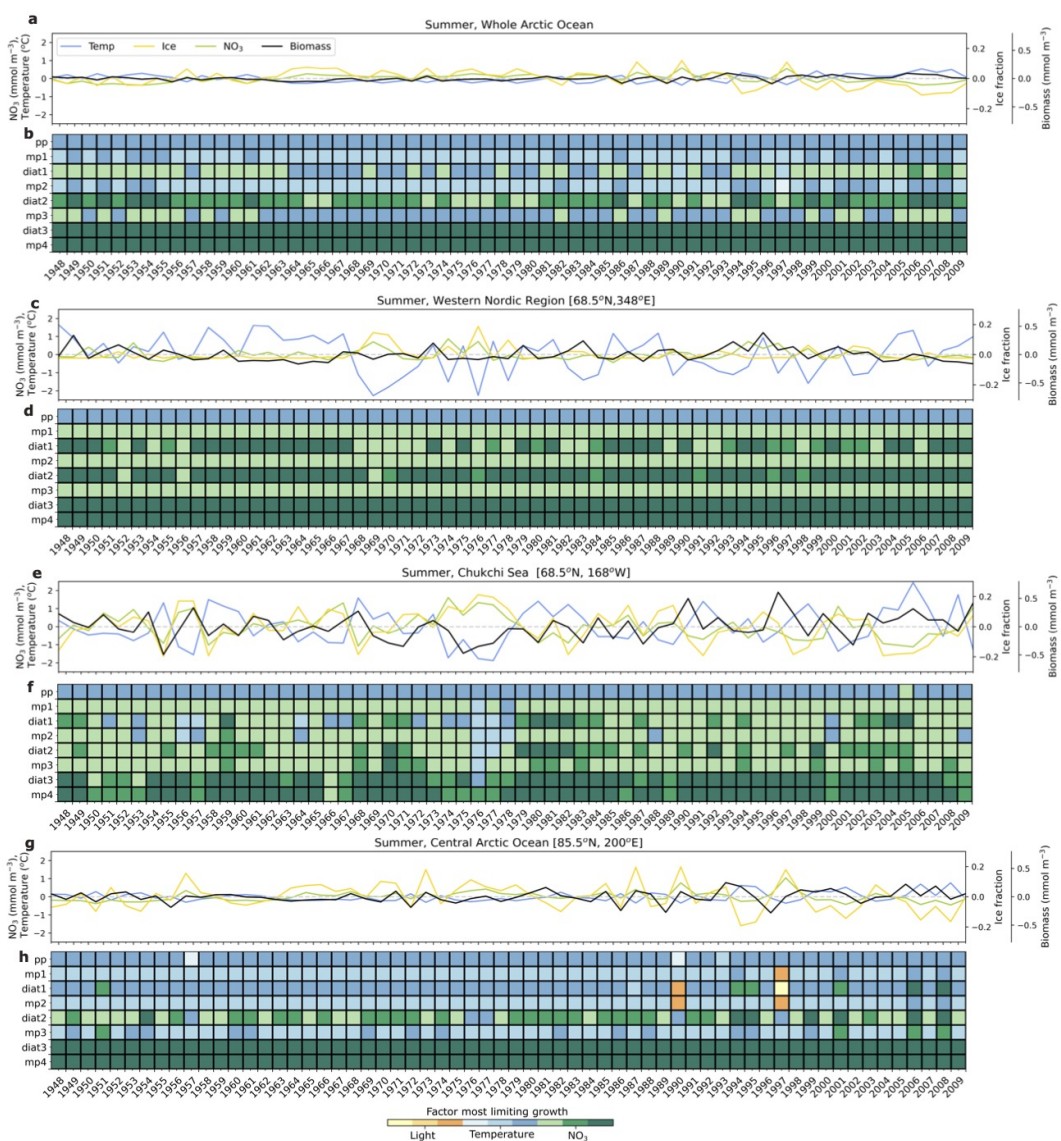

**Figure 8.** Summer anomalies from 1948-2009 in total phytoplankton biomass (mmol C m$^{-3}$; black line), nitrate (mmol N m$^{-3}$; green line), temperature ($^o$C; blue line), and sea ice fraction (yellow line) for the whole Arctic Ocean (a), and locations along the Western Nordic Seas [68.5$^o$N, 348$^o$E] (c), Chukchi Sea [68.5$^o$N, 168$^o$W] (e), and central Arctic Ocean [85.5$^o$N, 200$^o$E] (g). Anomalies were calculated relative to monthly averages of the 62 year (1948-2009) climatology. Phytoplankton growth limitation in summer is shown for each zone in b, d, f, h. The colors in b, d, f, and h indicate the factor that is most limiting (temperature=blue, light=yellow, nitrate=green), where darker shades of each color represent greater limitation. Nutrient and light limitation terms were computed as biomass-weighted vertical averages of the top 100m. Temperature limitations were estimated using activation energy values for each phytoplankton type.

fluctuations significantly affected small and medium-sized diatoms, modifying the interannual strength of nutrient limitation. The Chukchi Sea exhibited a temperature-driven positive correlation with biomass, primarily for small and medium-sized phytoplankton, with some transitioning from nutrient to temperature limitation during years of elevated nutrient concentrations. In both of these regions, years with elevated temperature and nutrient concentrations led to increased phytoplankton biomass for both small and large phytoplankton (Supp. Figs. S3 & S4). In the Central Arctic Ocean, substantial temperature, ice fraction, and nutrient anomalies created variability in the growth limitation of most small and medium-sized phytoplankton, with the largest phytoplankton consistently nutrient-limited. Here, increased phytoplankton biomass was consistent with years of elevated temperature, and lower ice fraction, even if nutrient concentrations were also lower (Supp. Figs. S3 & S4) reflecting the stronger sensitivity in the growth of phytoplankton to light availability from decreased ice fraction, than from changes in nutrient concentrations.

### 3.3.2   Phytoplankton community size structure

Interannual variations in ice fraction (Fig. 9e-h), temperature (Fig. 9i-l), and $NO_3$ concentrations (Fig. 9m-p) altered the slope of the size abundance relationship both regionally and seasonally. In most of the Arctic and across seasons, lower ice fraction and warmer temperatures led to more negative slopes, particularly along the ice edge in spring (Fig. 9 f,j). This response occurred because small phytoplankton became more abundant with greater light and warmer temperatures, similar to seasonal changes in size abundance slope (Supp. Fig. S3). This is consistent with past studies showing a change from larger diatoms to smaller functional types as the Arctic freshens (Li et al., 2009; Neeley et al., 2018). In contrast, years with more nitrate had higher size abundance slopes (Fig. 9m-p). In years with higher ice fraction and lower temperature, light and/or temperature strongly limited phytoplankton growth, leaving higher surface nitrate. The slope was consequently flatter due to lower biomass of small phytoplankton under these conditions.

Our results showed that interannual shifts in the size structure of phytoplankton communities were sensitive to fluctuations in growth limitation, consistent with previous studies emphasizing the sensitivity of small phytoplankton to changing Arctic conditions (Freyria et al., 2021; Wietz et al., 2021). Unlike more nutrient-limited systems at lower latitudes (e.g., Marañón et al., 2018), interannual variations in size structure were due primarily to changes in light and temperature, rather than nutrient supply.

### 3.3.3   Trophic fluxes in the plankton community

Interannual variability in ice fraction, temperature, and nutrient concentrations influenced the zooplankton mean trophic level. Zooplankton mean trophic levels were generally higher in winter, summer, and fall in years with low ice fraction and high temperatures in the Central Arctic Ocean (Fig. 10). In years with low ice fraction and higher temperature, small phytoplankton (Supp. Fig. S3) and microzooplankton were more abundant, such that the dominant trophic exchange was between smaller phytoplankton to herbivorous microzooplankton to mesozooplankton. The increased mesozooplankton trophic level was driven by a shift from mesozooplankton mainly feeding on large phytoplankton, to feeding on microzooplankton as well.

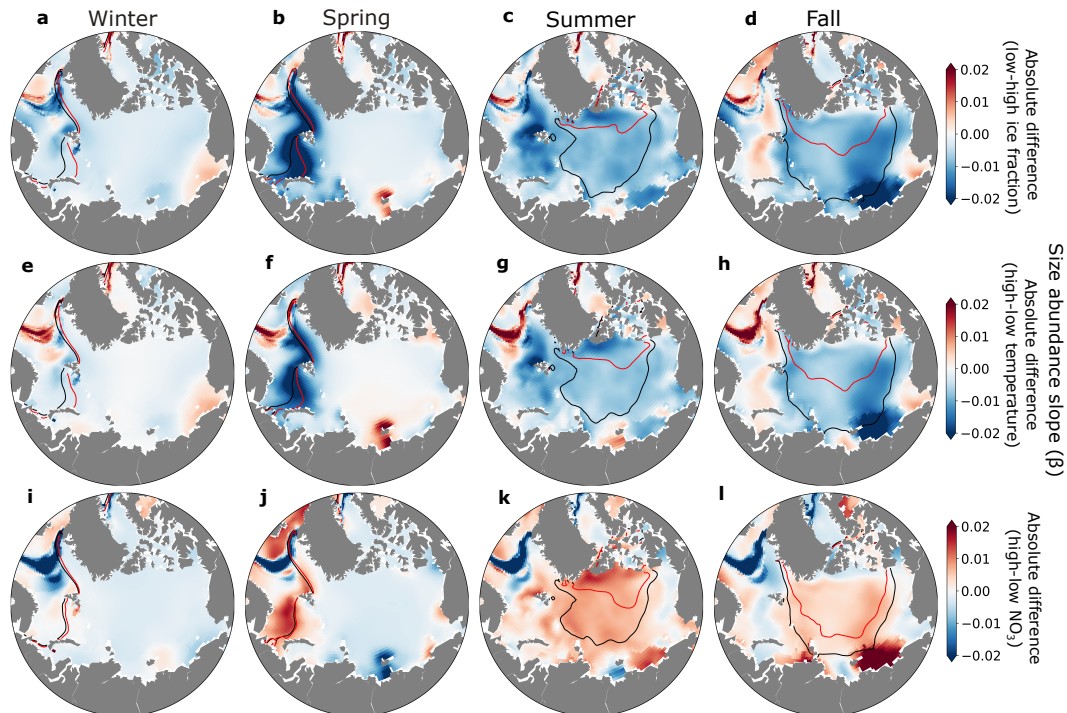

**Figure 9.** Absolute differences in size abundance slope between low and high ice (a-d), temperature (e-h), and NO$_3$ (i-l) years for winter (a,e,i,m), spring (b,f,j,n), summer (c,g,k,o), and fall (d,h,l,p). Black contour lines indicate the sea-ice extent in years with high ice fraction (a-d), low temperatures (e-h), and low nutrients (i-l). Red contour lines indicate sea-ice extent in years with low ice fraction (a-d), high temperature (e-h), and high nutrients (i-l).

Two distinct mechanisms of trophic change were evident in our analysis. In the central Arctic, the increase in zooplankton mean trophic level during years of elevated temperature, reduced ice fraction, and diminished NO$_3$ was caused by the earlier onset of phytoplankton and microzooplankton blooms, prompting mesozooplankton to graze on microzooplankton earlier in the year (Supp. Fig. S5). In contrast, during fall and winter, the inflow regions of the western Nordic Seas exhibited lower zooplankton mean trophic levels in years with higher temperatures and reduced ice coverage. These conditions were associated with overall declines in phytoplankton biomass (Supp. Figs. S3-S4), particularly among smaller phytoplankton (Supp. Fig. S6). This reduction in smaller phytoplankton led to a decline in microzooplankton, resulting in less grazing by mesozooplankton on microzooplankton. In the western Nordic Seas, the decreased phytoplankton biomass in fall and winter may have resulted from a chain reaction triggered by a strong phytoplankton bloom earlier in the season (e.g., spring, Supp. Figs. S3-S4), which quickly depleted available nutrients. In this typically light-limited region, reduced ice cover alleviated light limitation, allowing larger phytoplankton to utilize nutrients more effectively, leading to overall lower nutrient levels throughout the season.

These results align with observed regional differences in plankton dynamics throughout the Arctic due to earlier annual ice retreats and increased light availability (Song et al., 2021). They also emphasize how planktonic organisms exhibit varied

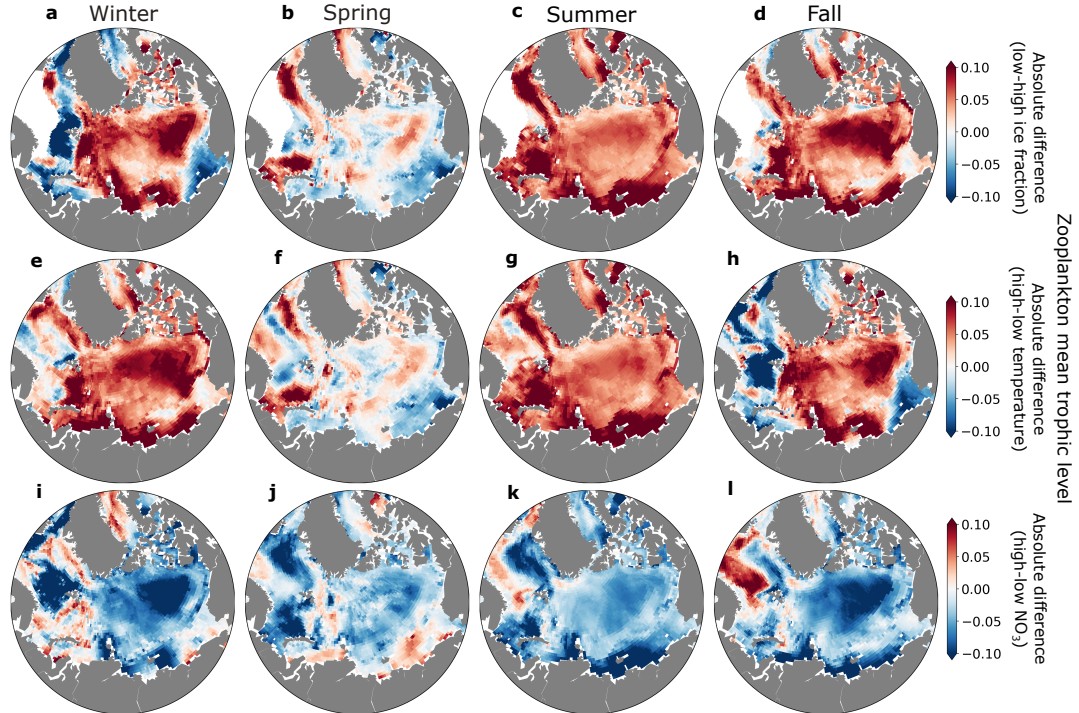

**Figure 10.** Absolute differences in zooplankton mean trophic level between low and high ice (a-d), temperature (e-h), and NO$_3$ (i-l) years for winter (a,e,i), spring (b,f,j), summer (c,g,k), and fall (d,h,l).

responses to the same environmental changes, consistent with previous studies investigating the impacts of warming across trophic levels and functional groups within an ecological community (Edwards and Richardson, 2004).

### 3.3.4 Fisheries production

Estimated annual fisheries production in the Arctic Ocean averaged over the last 20 years (1990-2009) of the hindcast simulation exhibited considerable regional variability, which was closely linked to changes in mesozooplankton biomass and carbon export to the benthos. Energy available from the flux of detritus to the benthos was highest in the continental shelves and shallower regions and very low in the deep Arctic Ocean due to the strong remineralization over the the deeper water column there (Fig. 11a). Energy available from mesozooplankton production was relatively low in the Central Arctic Ocean where

sea ice and temperature placed strong bottom-up constraints on system primary productivity (Fig. 11c). The combined (Fig. 11c) view showed that estimated fish production was particularly high in the Barents, Nordic, and Chukchi Seas (Fig. 11c). Pelagic-benthic coupling describes the connection between surface-water production and seafloor habitats via energy, nutrient, and mass exchange, and is considered tighter in areas with sea ice (Zhulay et al., 2023). Shallow Arctic shelves are characterized by tight pelagic-benthic coupling due to low grazing in the water column during blooms (Grebmeier et al., 1988, 1989;

Renaud et al., 2008; Tamelander et al., 2008), resulting in large export of organic matter from surface layers to the seafloor and benthos, especially when production exceeds zooplankton consumption (Tamelander et al., 2006).

Years with lower ice fraction and warmer temperature generally supported higher fish production (Fig. 11f,i). This increase was primarily due to increased primary and mesozooplankton production (Supp. Figs. S4,S3 & S5) in regions where light and temperature limitation of phytoplankton growth was reduced (Fig. 5). The flux of detritus to the benthos also increased

where the surface production increased, but this source of energy for fish production was most important in shallow, coastal areas (Fig. 11d,g), and was generally less than fish production due to mesozooplankton in the water column (Fig. 11e,h). An exception to these interannual changes was the Nordic and Chukchi Seas. Here, estimated fish production decreased in low ice and warm years, likely because primary and zooplankton production in these regions tended to be limited by nutrients to a greater extent than by light or temperature. These areas showed higher fish production in high nitrate years (Fig. 11l). While

our analysis of fish production in the Arctic Ocean compares years with contrasting environmental conditions in the historic simulations, modeling studies suggest that by the year 2100, fish catch in the Arctic Ocean could increase (Tai et al., 2019). The warm and low ice periods in our historic model simulations may provide an analog for a future Arctic Ocean that is warmer with significantly less ice.

## 4   Model study limitations

While this study utilized a state-of-the-art, intermediate complexity plankton community model embedded a three dimensional simulation of ocean circulation and biogeochemical cycles, there are important model limitations that we discuss here.

One significant limitation of this study was the absence of sea ice algae within the ice-ocean-biogeochemistry model. Sea ice algae play crucial roles in shaping sea-ice associated ecosystems and biogeochemistry (Kohlbach et al., 2016), and their absence in our model may lead to an underestimation of the contributions of ice-associated production to benthic and pelagic ecosys-

tems during periods of high sympagic activity. However, their influence at regional and global scales remains unclear, in part due to the limited spatial and temporal coverage of observations (Hayashida et al., 2021). Additionally, there are significant uncertainties and omissions in the representation of zooplankton, their grazing, and population dynamics in MARBL-SPECTRA. Zooplankton grazing (including food web structure, grazing parameters, and functional responses) is one of the largest sources of uncertainty in ocean biogeochemical models (Rohr et al., 2023; Hansen et al., 1997; Gentleman and Neuheimer, 2008), and

in this model, we do not resolve key zooplankton life histories or migration dynamics such as dormancy or diapause, which are important traits that allow organisms to survive unfavorable environmental conditions in the Arctic (Baumgartner and Tarrant, 2017). The omission of these traits may lead to an incomplete representation of zooplankton population dynamics and their seasonal availability as prey. Consequently, this limitation influenced the underestimation of mesozooplankton biomass (Fig. 4o), potentially influencing our assessments of trophic interactions and fisheries production.

We utilized the empirical model of Stock et al. (2017) to estimate fisheries production in the Arctic region, but this model may not capture important aspects of fish production in the Arctic. For example, the model does not explicitly account for fish feeding under the ice, a significant process within the Arctic (Kohlbach et al., 2017). Additionally, the empirical relationship

does not differentiate between fish life stages or sizes within the selected trophic level, limiting the results to an estimate of overall fisheries production.

The hindcast simulations employed the CORE-II dataset, a commonly-used approached which allows for comparison across models and the integration of ocean-ice models without fully coupling to atmospheric Global Circulation Models (Griffies et al., 2009). CORE-II relies on NCEP-NCAR reanalysis and satellite data when available(Large and Yeager, 2009). When data coverage were insufficient, CORE-II forcing fields for some variables are repeating seasonal cycles, rather than complete, interannually varying records. Additionally, CORE-II is not optimized for the Arctic due to the scarcity of observations there. Simulations using CORE-II have found smaller than observed multi-model-mean sea ice extent, particularly in the summer (Wang et al., 2016a, b).

## 5 Conclusion

This study used a simulation of plankton communities, biogeochemical cycles, ocean circulation, and sea ice to better understand how seasonal to interannual changes in the environment influence phytoplankton physiology, plankton community structure and trophic dynamics, and fish production in the Arctic Ocean, an undersampled portion of the global ocean.

The growth of model phytoplankton was primarily limited in winter, spring, and fall by light, but in summer, the growth of smaller and larger phytoplankton was most limited by temperature and nutrient availability, respectively. The dominant trophic pathway in summer was from phytoplankton to herbivorous zooplankton, such that the average trophic position of model zooplankton was lowest in the summer growing season. Changes in the slope of the size abundance relationship in the model were driven to a large extent by changes in abundance of smaller phytoplankton, not larger phytoplankton, as has often been found in lower-latitude systems.

On interannual timescales, the model indicated that changes in phytoplankton biomass in summer were most sensitive to variations in temperature and ice cover in the pelagic regions of the Arctic. This sensitivity underscores the vulnerability of these regions to climate fluctuations, with potential repercussions for the entire marine ecosystem. In contrast, inflow and outflow regions of the Nordic, Barents and Chukchi Seas exhibited higher sensitivities in phytoplankton biomass to nutrient availability. This suggests that the nutrient dynamics in these areas play a crucial role in shaping phytoplankton abundance.

The size abundance relationship slopes were steeper in summers with increased temperature, and reduced ice fraction, reflecting a phytoplankton community enriched in smaller species that have higher affinity for light than do larger phytoplankton. This shift towards smaller species caused higher mean zooplankton trophic positions, indicative of increased zooplankton carnivory. These changes were due, in part, to changes in phenology, with blooms occurring early in the year.

We estimated greater fish production in years with lower ice and warmer temperatures, due to higher primary and mesozooplankton production in those years. This underscores the intricate connections between climate, plankton dynamics, and fishery resources. The implications extend beyond ecological considerations to economic and societal aspects, emphasizing the need for adaptive strategies in the face of ongoing climate change for sustainable fisheries and ecosystem health.

Overall, our model simulations provide mechanistic insights into the seasonal to interannual dynamics of Arctic Ocean marine ecosystems and highlight the need for continued monitoring and modeling efforts to understand and predict the impacts of climate change on these ecosystems. While the ecological, physical, and biogeochemical models include many simplifications that reflect a balance between realism and computational feasibility, they provide a quantitative tool to examine regional, seasonal, and interannual patterns in key ecosystem processes, as well as the environmental and biotic mechanisms that shape them.

## 5.1 Code and data availability

Data from the MARBL-SPECTRA simulations performed for this study are available at: https://doi.org/10.6075/J0NK3F6D. All codes used for analysis are available at: https://github.com/gabyneg/arctic_analysis/

## 5.2 Author contribution

GNG conducted the numerical experiments, analysed and visualised the results, and led the writing process; JYL & ADB contributed to the experimental design, to the results analysis, and to the writing process. All authors (GNG, JYL, CMP, MM & ADB) discussed and refined the text and contributed to the interpretation of results.

## 5.3 Competing interest

The authors declare that they have no conflict of interest.

## 6 Acknowledgements

We acknowledge high-performance computing support from Cheyenne (doi:10.5065/D6RX99HX) provided by NCAR's Computational and Information Systems Laboratory, sponsored by the National Science Foundation (NSF). This manuscript is based upon work supported by the National Center for Atmospheric Research, which is a major facility sponsored by the National Science Foundation under Cooperative Agreement No. 1852977, and the National Science Foundation Graduate Research Fellowship under Grant No. (DGE-2038238 and DGE-1650112). MM is grateful to NASA (IDS-#19-0113) and NSF (OPP-192292 and OCE-2049294) for financial support. Any opinions, findings, and conclusions or recommendations expressed in this material are those of the author(s) and do not necessarily reflect the views of the National Science Foundation. We also gratefully acknowledge comments from Matthew C. Long and Charles Stock on this manuscript.

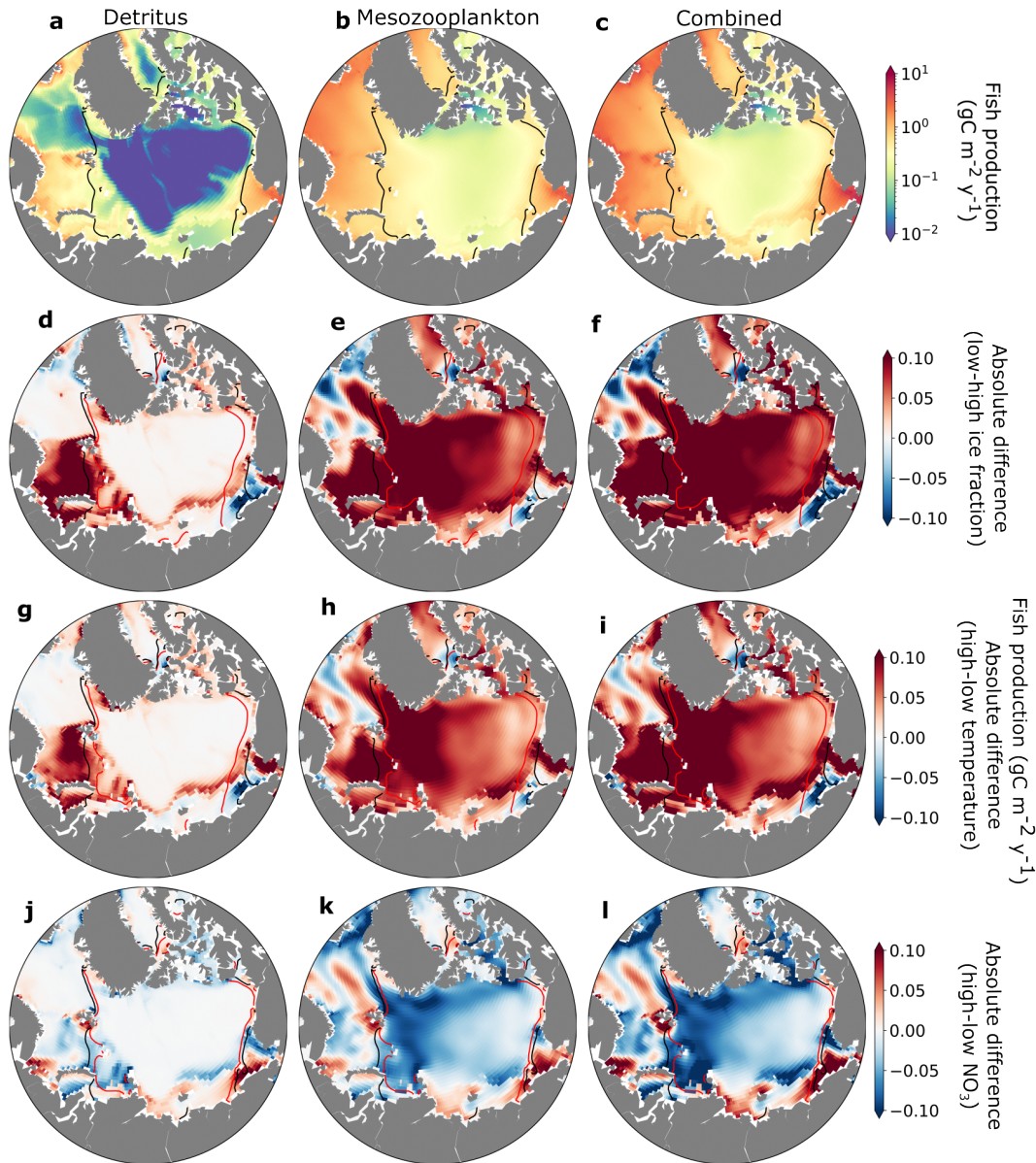

**Figure 11.** Estimates of annual fish production due to fluxes of detritus to the benthos (a), mesozooplankton production in the water column (b), and both sources of energy combined (c) zones (gC m$^{-2}$ y$^{-1}$). Absolute fish production differences (gC m$^{-2}$ y$^{-1}$) between low and high ice fraction (d,e,f), temperature (g,h,i), and NO$_3$ (j,k,l) years, separated by column to the source of the production (left column: detritus, middle column: mesozooplankton production, right column: combined). Years of high and low temperatures, ice, and NO$_3$ concentrations were selected corresponding to the $90^{th}$ and $10^{th}$ percentiles, respectively, based on the mean climatology. Black contour lines indicate the sea-ice extent in years with high ice fraction (d-f), low temperatures (g-i), and low nutrients (j-l). Red contour lines indicate sea-ice extent in years with low ice fraction (d-f), high temperature (g-i), high nutrients (j-l)

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
