# Peer review of "Changes in Arctic Ocean plankton community structure and trophic dynamics on seasonal to interannual timescales"

_EGUsphere, 2024_

## Author Comment (AC1)

**Responses to Reviewer 1:**

**Minor comments:**

**L120-125 It would be useful to also detail how sea-ice influences light limitation in the model. Is NPP permissible under sea ice? If so, does this depend on thickness, snow cover etc?**

- Thank you for your comment. Yes, NPP occurs below sea ice in the water column. In the model, sea ice thickness determines the light entering the surface layer below the ice. Our model used the approach developed by Long et al. (2015). The model does not use the grid-cell mean irradiance to compute phytoplankton light limitation terms. Instead, it computes light limitation across different categories of ice thickness present within a grid cell, then averages the limitation terms across the grid cell. This approach acknowledges that light penetration varies significantly with the thickness of the sea ice, providing a more nuanced and accurate representation of light availability under the ice.
- In the revised manuscript, we added the following text (L127-L130): "Considerable spatial heterogeneity exists in sea ice thickness, which affects light available for phytoplankton growth below sea ice. Following the approach of Long et al (2015), our model calculates phytoplankton growth limitation terms for the distribution of sea ice thicknesses present within the model grid cell, and then averages these values over the grid cell to estimate the average light limitation in the grid cell."

**L161-163. I don't understand this. Do the authors mean that atmospheric greenhouse gas concentrations are held constant in these simulations so the impact of acidification for example is not simulated? The prescribed CORE-II fluxes (e.g. heat and freshwater fluxes) are presumably affected by anthropogenic climate change. Or have these forcings been modified in some way?**

- Thank you for your comment. The COREv2 dataset encompasses a comprehensive set of air-sea fluxes, including momentum, heat, and freshwater, spanning from 1948 to 2009 at monthly resolution. The historical climate signal is indeed embedded within the atmospheric reanalysis data and satellite-derived flux measurements used to force the model. However, because the seasonal-to-interannual variability in the CORE-II forcing is large compared with the longer-term climate signal, we considered that our simulations are more appropriate for studying seasonal to interannual variations. Therefore, we focused on the seasonal and interannual changes in light, temperature, and nutrients. In the revised manuscript, we removed the confusing sentence "It is not a simulation forced by increasing greenhouse gas emissions, and as such, our analyses focus on seasonal to interannual rather than decadal-to-centennial changes forced by anthropogenic greenhouse gas emissions." and replaced it with, (L167-L169): "Though the CORE-II

forcing captures the historical climate signal, we instead focus on seasonal-to-interannual variability because it is the larger signal compared to climate change during this time period."

**L202 Is NO3 the only limiting nutrient in the Arctic domain or the only nutrient that is assessed?**

- The model simulates transformation and transport of nitrogen, phosphorus, iron, and silicate. We identified the most limiting nutrient for each phytoplankton type (following Leibig's Law of the Minimum (Equation 2 shown below)) and used this information to compare nutrient limitation with limitations imposed by light and temperature (as shown in Figure 5 of our manuscript). In the Arctic Ocean model, nitrate was the predominant limiting nutrient for phytoplankton growth in most places and times (see figure below), consistent with other modeling studies (Manizza et al. 2023). Because of this, we elected to focus primarily on nitrate.
  Equation 2: $\gamma_i{}^N = min(N_{N,i}^{lim}, \ N_{P,i}^{lim}, N_{Fe,i}^{lim}, N_{Si,i}^{lim})$

- In the revised manuscript, we added a supplemental figure showing that nitrate is the most widely limiting nutrient in this region across all groups of phytoplankton (see figure below). This figure also shows that iron limitation is important in summer in the subpolar portions of the North Atlantic Ocean. Additionally, we updated the description in Figure 5 to include "Nitrate was the most limiting nutrient for phytoplankton growth for most regions and seasons (Supp. Fig. S2)."

[Figure]

Figure S2. Most limiting nutrient for each phytoplankton group in the Arctic Ocean. The most limiting nutrients (Phosphate (purple), Iron (yellow), Nitrate (green), and Silica (red)) for each phytoplankton type were averaged over the three months in each season, averaged over 1990-2009. White regions represent areas where no nutrient is limiting. Silica was only considered for diatoms.

**L270-272. It is difficult to reconcile this text with what I can interpret from Figure 3. Few if any of the subregions seem to exhibit peaks in observed and simulated chlorophyll in the same month. In the East Siberian Sea, Chukchi Sea and Beaufort Sea the simulated peak appears to be 2-3 months later.**

- In response to your comment as well as similar concerns from Reviewers 2 and 3, we have modified the satellite and model Chl comparisons, and now use a satellite ocean color algorithm to estimate chlorophyll tailored for the Arctic Ocean developed by Lewis and Arrigo (2020; https://doi.org/10.1029/2019JC015706). The Lewis and Arrigo approach uses a large bio-optical database from the Arctic Ocean to generate estimates of chlorophyll specific to the Arctic Ocean. We have recreated Figures 2 and 3 with this satellite chlorophyll estimate.

[Figure]

Figure 2 (*just the chlorophyll portion*). Annual average model (j) and satellite (k) surface chlorophyll, and difference between model minus satellite ($log_{10}$ mg Chl $m^{-3}$).

[Figure]

Figure 1. Map of the Arctic Ocean divided into ten sections included in the analysis, following Lewis & Arrigo (2020) regional mask. Additionally, symbols represent three grid cells selected throughout the Arctic Ocean for growth limitation analysis. The cross symbol shows the Western Nordic Seas ($68.5^o$N, $348^o$E), the star symbol shows the central Arctic Ocean location ($85.5^o$N, $200^o$E), and the triangle symbol shows the Chukchi Sea location ($68.5^o$N, $168^o$W).

[Figure]

Figure 3. Modeled and satellite estimates of seasonal variability in surface chlorophyll. The solid blue lines depict the modeled monthly-averaged chlorophyll at the surface layer (10 m), while the dashed blue lines represent the satellite-derived (Lewis & Arrigo 2020) surface chlorophyll. Additionally, the black line shows modeled monthly-averaged ice fraction, and the thin red line represents the average photosynthetically active radiation (PAR) over the surface layer (10 m) (W $m^{-2}$). Seasonal cycles are displayed for nine different regions: the Chukchi Sea (a), Barents Sea (b), Siberian Sea (c), Laptev Sea (d),  Kara Sea (e), Beaufort Sea (f), Baffin Bay (g), Canadian Archipelago (h), and Nordic Sea (i).

- We have also modified the text as follows:
- L260-L264: "MARBL-SPECTRA generally underestimated surface chlorophyll along coastal waters above the Russian continental shelves compared to satellite-based chlorophyll estimates using an ocean color algorithm tailored for the Arctic Ocean developed by Lewis et al. 2020 (Fig. 2l). This underestimation can be attributed to inaccuracies of the satellite estimates from the atmospheric correction scheme, sensor calibration, or bio-optical algorithms, which were not optimized to account for the presence of colored dissolved organic matter (CDOM) in coastal waters (Siegel et al.,2013, 2002; Mustapha et al. 2012)."

- L273- 277: "Comparison between satellite and model chlorophyll is difficult due to known challenges of remote sensing in the Arctic Ocean, including but not limited to clouds, sea ice, and organic matter in the water column (Li et al. 2024, Gregg and Casey, 2007, Mikelsons and Wang, 2019). However, we further assessed the performance of MARBL-SPECTRA by comparing the seasonality of surface chlorophyll in different regions of the Arctic Ocean with satellite chlorophyll estimates tailored to the Arctic Ocean (Lewis and Arrigo, 2020) (Fig. 3).
- L279-L290: "With the exception of the Chukchi (Fig 3a) and Barents Seas (Fig 3b), model and satellite chlorophyll magnitudes were qualitatively similar. The satellite and model seasonal phenology of chlorophyll were similar in some regions (e.g., the Nordic Sea (Fig. 3i)) but shifted in others (e.g., Baffin Bay (Fig. 3g), Barents Sea (Fig. 3b))) due to temporal discrepancies between model and remotely sensed Arctic Ocean in timing of sea ice retreat. MARBL-SPECTRA simulated a summer peak in chlorophyll during July in the Siberian (Fig.3c), Laptev (Fig. 3d), Kara (Fig. 3e) Seas and Canadian Archipelago (Fig. 3h), coinciding with the highest average photosynthetically active radiation (PAR) over the surface layer and a rapid decrease in sea ice fraction. In the Barents Sea (Fig. 3b) and Baffin Bay (Fig. 3g), MARBL-SPECTRA simulated a peak in chlorophyll concentrations of lower magnitude than the satellite estimate, and with a month delay. Comparing the Central Arctic region with satellite-based estimates was challenging due to the limited chlorophyll information available, as this area remains mostly covered by ice throughout the year."

**While in the Barents Sea the seasonal cycles of observed and simulated chlorophyll almost appear to be anticorrelated. What explains the winter peak in observed chlorophyll in the Barents Sea? Presumably there is insufficient light availability to sustain this? Is this an artifact of variable observational coverage? In which case it might be best to only do pairwise comparisons of models and obs.**

- In our updated comparison between model chlorophyll and satellite chlorophyll, estimated using an algorithm tailored to the Arctic Ocean (Lewis and Arrigo, 2020), we observe that the previously noted peak in chlorophyll during the winter months in the Barents Sea is no longer present. This suggests that the peak may have been due to biases in the earlier satellite product we used.

**Figure 5. I'm surprised that in the Central Arctic light limitation isn't more extensive in summer. It would be useful to add an evaluation of simulated sea ice extent/thickness. Maybe in Figure 2. Overestimation of seasonal sea ice variability might help explain this and as the authors mention in their discussion, this is an issue that has been previously identified with CORE-II forced simulations.**

- Figure 5 shows the most limiting factor. This does not mean that phytoplankton growth in the Central Arctic isn't inhibited by light, only that nutrient limitation is stronger. However, we appreciate your suggestion to include model data on sea ice fraction, and have included it in the revised manuscript.

[Figure]

Figure S1. Ecosystem ice fraction in winter (a, December-February), spring (b; March-May), summer (c; June-August), and fall (d; September-November), averaged over 1990-2009.

**L365 does "added" just mean simulated here?**

- Thank you for catching that, yes, we meant that larger phytoplankton were increased seasonally in simulations and modified the text to say "simulated to increase".

**Figure 8 is quite difficult to interpret. I suggest avoiding the repetition of labels to allow you to increase the figure size. Why does summer sea ice not appear to be declining? Is this a shortcoming of the simulation setup as mentioned in line 478?**

-   Thank you for your comment. We updated Figure 8 by removing some of the yearly labels and making the y axes on the time series plots consistent so that it provides more clarity.

[Figure]

Figure 8. Summer anomalies from 1948-2009 in total phytoplankton biomass (mmol C m−₃; black line), nitrate (mmol N m−₃; green line), temperature (oC; blue line), and sea ice fraction (yellow line) for the whole Arctic Ocean (a), and locations along the Western Nordic Seas [68.5oN, 348oE] (c), Chukchi Sea [68.5oN, 168oW] (e), and central Arctic

Ocean [85.5∘N, 200∘E] (g). Anomalies were calculated relative to monthly averages of the 62 year (1948-2009) climatology. Phytoplankton growth limitation in summer is shown for each zone in b, d, f, h. The colors in b, d, f, and h indicate the factor that is most limiting (temperature=blue, light=yellow, nitrate=green), where darker shades of each color represent greater limitation. Nutrient and light limitation terms were computed as biomass-weighted vertical averages of the top 100m. Temperature limitations were estimated using activation energy values for each phytoplankton type.

- The hindcast simulation showed modest reductions in sea ice extent, but these changes were not uniform across all specific regions and time periods. This is a known discrepancy with Arctic CORE-II forced models (Wang et al. 2016a, 2016b). This model limitation is why our analysis emphasized seasonal and interannual variability rather than focusing solely on a long-term decline in sea ice and associated changes in the plankton.

**L433-437 It's not entirely clear to me why these regions are behaving differently. Can the authors expand a little here? Under diminished NO3 one would typically expect fewer large phytoplankton. Why is this not occurring in the Nordic Seas?**

- Thank you, we have incorporated a more detailed description in this section to explain this behavior. This region in the western Nordic Sea is typically light-limited throughout the year. Therefore, reductions in ice cover decrease light limitation, allowing larger phytoplankton to utilize available nutrient resources more effectively. The observed decrease in nutrients may be due to an earlier phytoplankton bloom, which depletes nutrient concentrations earlier in the season. This pattern results in lower nutrient levels when averaged over the entire season.
- We will modify the text to how say:

    L454-L468: "Two distinct mechanisms of trophic change were evident in our analysis. In the central Arctic, the increase in zooplankton mean trophic level during years of elevated temperature, reduced ice fraction, and diminished NO$_3$ was caused by the earlier onset of phytoplankton and microzooplankton blooms, prompting mesozooplankton to graze on microzooplankton earlier in the year (Supp. Fig. S7). In contrast, during fall and winter, the inflow regions of the western Nordic Seas exhibited lower zooplankton mean trophic levels in years with higher temperatures and reduced ice coverage. These conditions were associated with overall declines in phytoplankton biomass (Supp. Figs. S3-S4), particularly among smaller phytoplankton (Supp. Fig. S6). This reduction in smaller phytoplankton led to a decline in microzooplankton, resulting in less grazing by mesozooplankton on microzooplankton. In the western Nordic Seas, the decreased phytoplankton biomass in fall and winter may have resulted from a chain reaction triggered by a strong phytoplankton bloom earlier in the season (e.g., spring, Supp. Figs. S3-S4), which quickly depleted available nutrients. In this typically light-limited region, reduced ice cover alleviated light limitation, allowing larger phytoplankton to utilize nutrients more effectively, leading to overall lower nutrient levels throughout the season.

These results align with observed regional differences in plankton dynamics throughout the Arctic due to earlier annual ice retreats and increased light availability (Song et al. 2021). They also emphasize how planktonic organisms exhibit varied responses to the same environmental changes, consistent with previous studies investigating the impacts of warming across trophic levels and functional groups within an ecological community (Edwards et al. 2004)."

**References:**

Edwards, M., & Richardson, A. J. (2004). Impact of climate change on marine pelagic phenology and trophic mismatch. *Nature*, *430*(7002), 881-884.

Gregg, W. W., & Casey, N. W. (2007). Sampling biases in MODIS and SeaWiFS ocean chlorophyll data. *Remote Sensing of Environment*, *111*(1), 25-35.

Lewis, K. M., & Arrigo, K. R. (2020). Ocean color algorithms for estimating chlorophyll a, CDOM absorption, and particle backscattering in the Arctic Ocean. *Journal of Geophysical Research: Oceans*, *125*(6), e2019JC015706.

Li, J., Matsuoka, A., Pang, X., Massicotte, P., & Babin, M. (2024). Performance of Algorithms for Retrieving Chlorophyll a Concentrations in the Arctic Ocean: Impact on Primary Production Estimates. *Remote Sensing*, *16*(5), 892.

Long, M. C., Lindsay, K., & Holland, M. M. (2015). Modeling photosynthesis in sea ice-covered waters. *Journal of Advances in Modeling Earth Systems*, *7*(3), 1189-1206.

Manizza, M., Carroll, D., Menemenlis, D., Zhang, H., & Miller, C. E. (2023). Modeling the recent changes of phytoplankton blooms dynamics in the Arctic Ocean. *Journal of Geophysical Research: Oceans*, *128*(6), e2022JC019152.

Mikelsons, K., & Wang, M. (2019). Optimal satellite orbit configuration for global ocean color product coverage. *Optics Express*, *27*(8), A445-A457.

Mustapha, S. B., Belanger, S., & Larouche, P. (2012). Evaluation of ocean color algorithms in the southeastern Beaufort Sea, Canadian Arctic: New parameterization using SeaWiFS, MODIS, and MERIS spectral bands. *Canadian Journal of Remote Sensing*, *38*(5), 535-556.

Siegel, D. A., Maritorena, S., Nelson, N. B., Hansell, D. A., & Lorenzi-Kayser, M. (2002). Global distribution and dynamics of colored dissolved and detrital organic materials. *Journal of Geophysical Research: Oceans*, *107*(C12), 21-1.

Siegel, D. A., Behrenfeld, M. J., Maritorena, S., McClain, C. R., Antoine, D., Bailey, S. W., ... & Yoder, J. A. (2013). Regional to global assessments of phytoplankton dynamics from the SeaWiFS mission. *Remote Sensing of Environment*, *135*, 77-91.

Song, H., Ji, R., Jin, M., Li, Y., Feng, Z., Varpe, Ø., & Davis, C. S. (2021). Strong and regionally distinct links between ice-retreat timing and phytoplankton production in the Arctic Ocean. *Limnology and Oceanography*, *66*(6), 2498-2508.

Wang, Q., Ilicak, M., Gerdes, R., Drange, H., Aksenov, Y., Bailey, D.A., Bentsen, M., Biastoch, A., Bozec, A., Böning, C., et al., 2016a. An assessment of the arctic ocean in a suite of interannual core-ii simulations. part i: Sea ice and solid freshwater. Ocean Modelling 99, 110–132.

Wang, Q., Ilicak, M., Gerdes, R., Drange, H., Aksenov, Y., Bailey, D.A., Bentsen, M., Biastoch, A., Bozec, A., Böning, C., et al., 2016b. An assessment of the arctic ocean in a suite of interannual core-ii simulations. part ii: Liquid freshwater. Ocean Modelling 99, 86–109.

---

## Author Comment (AC2)

**Responses to Reviewer 2:**

**Minor comments:**

**Chl validation: Here, modeled Chl is compared to satellite-derived Chl using a global algorithm. The Arctic has relatively unique ocean optics, and global ocean color algorithms do not typically replicate Arctic Chl well, so I'd recommend using an Arctic-specific algorithm. This will likely lead to a reduction in satellite-derived Chl, making this comparison look far better, and will also likely shift the seasonality of the phytoplankton blooms earlier.**

- Thank you for your comment, which is similar to Reviewer 1's concerns about the comparison with satellite chlorophyll estimates. In the revised paper, we used chlorophyll estimated with an algorithm tailored to the Arctic Ocean (Lewis and Arrigo, 2020; https://doi.org/10.1029/2019JC015706). Please see our response to Reviewer 1 for the new figures and a detailed discussion of model-satellite chlorophyll comparison.

**Nutrients: When nutrient limitation is discussed, does this refer exclusively to NO3 limitation? I imagine that diatoms are limited by Si at least seasonally or in some parts of the Arctic. A little more clarity about what nutrients limit phytoplankton growth would be appreciated.**

- Thank you for your comment about limiting nutrients, which is similar to a comment by Reviewer 1. Nitrate is the limiting nutrient in most parts of the Arctic Ocean, most of the time for most taxa. In the revised manuscript, we added a figure that shows this, and provided language to make this clearer. Please see the detailed response to Reviewer 1 for figure and text changes.
- For Figure 5, we aimed for simplicity by comparing growth limitation by nutrients, light, and temperature, but see how this introduced uncertainty. By "nutrients" in this figure, we mean any nutrient, but this is usually nitrate (see Figure S2 in response to Reviewer 1). In a revised manuscript, we will add the following text to make this clearer:

    L306-308: "In Figure 5, we assess whether phytoplankton growth was most limited by light, temperature, or nutrients. In this case, nutrients can refer to limitation by nitrate, phosphate, silicate, or iron, but in practice in the model, phytoplankton growth is most often limited by nitrate (Supp. Fig. S2)."

**Results and Discussion overall: I think this section would benefit from greater contextualization with/ comparison to previous studies – perhaps a few sentences in each section.**

In a revised manuscript, we incorporated greater contextualization with comparison to previous studies. We added the suggested text below, in response to individual points/questions.

**For example, you describe how phytoplankton biomass shifts from largely dominated by diatoms to dominated by smaller functional types as nutrients are drawn down. This is common in global oceans, but has it been observed (or found in other model configurations) in the Arctic? What about the seasonal succession of zooplankton you observe?**

- Thank you for your comment, we included the following in our discussion:
    - L349-351: "The model indicated a pronounced seasonal shift from large to small phytoplankton driven by the seasonal reduction of surface nitrate, which is largely consistent with observations from the region (Ardyna et al. 2017, Ardyna et al. 2011, Ardyna and Arrigo, 2020, Tremblay et al. 2009, Usov et al. 2024)."
- Regarding zooplankton seasonal succession, in our manuscript we included:
    - L351-354: "The seasonal succession of zooplankton is significantly influenced by the size structure of the phytoplankton, consistent with Usov et al. (2024), who found that distinct seasonal groups of phytoplankton and zooplankton in the Chupa Inlet (White Sea) are interconnected, with smaller species playing a larger role in summer and autumn, enhancing trophic coupling throughout the seasonal cycle."

-

**Similarly, discussion of the predominant limitation terms for phytoplankton growth in other models (or in CESM-1 with Krumhardt et al., 2020, for example) would be valuable.**

- The Krumhardt et al (2020) paper evaluates the correlation between NPP variability with variations in the light, temperature, and nutrient limitation terms, with the aim of elucidating the factors that influence the potential predictability of NPP. This is distinct from identifying the most limiting factor for phytoplankton growth. Additionally, those analyses were done on annually averaged data and did not evaluate seasonal changes in limiting factors, which we think is more appropriate for understanding the factors limiting phytoplankton growth, particularly in highly seasonal seas such as the Arctic. We believe that the analysis of seasonal limitation factors is one of the major contributions of our manuscript. Still, in consideration of this suggestion, we added the following text to the manuscript: L325- 328: "Our results concerning factors limiting phytoplankton growth are broadly consistent with previous modeling studies that have shown that temperature and light strongly limit phytoplankton growth in the Arctic Ocean (Steinacher et al. 2010; Krumhardt et al. 2020), however we provide additional context on seasonal changes and how limiting factors vary across phytoplankton sizes."

**For the fisheries production results, it might be useful to look at other modeled estimates of fisheries changes in the future (e.g. Tai et al., 2019). While this model is only run until 2009, your results about how biomass changes under low sea ice and warmer ocean temperatures suggest a more productive Arctic in the future. Is that consistent with other model findings? A few sentences about these results will better allow readers to assess which of your findings are new contributions and which are consistent with other observational studies or previous modeling studies, giving us confidence in this model configuration.**

- Tai et al. (2019) projected that fish catch in the Canadian Arctic may increase by 2100 under high emissions scenario (RCP8.5). In a revised manuscript, we added the following sentences: L489-493: "While our analysis of fish production in the Arctic Ocean compares years with contrasting environmental conditions in the historic simulations, these results align with observations of fish physiology in the Kitikmeot region of the Canadian Arctic, where years with earlier ice breakup, as observed by both Inuit fishers and biophysical indicators, showed positive effects on Arctic Char quality as reflected by both fish condition and lipid content (Falardeau et al., 2022). Other studies suggest there could be an increase in fish catch and economic value potential (Tai et al., 2019). Conversely, some studies exploring the ongoing impacts of Arctic warming predict that by the end of the century, Arctic fish species like Arctic Cod may decline due to rising ocean temperatures and shifts in habitat and migration patterns (Florko et al., 2021; Steiner et al., 2019)."

**References:**
Ardyna, M., Babin, M., Devred, E., Forest, A., Gosselin, M., Raimbault, P., & Tremblay, J. É. (2017). Shelf-basin gradients shape ecological phytoplankton niches and community composition in the coastal Arctic Ocean (Beaufort Sea). *Limnology and oceanography*, *62*(5), 2113-2132.

Ardyna, M., Gosselin, M., Michel, C., Poulin, M., & Tremblay, J. É. (2011). Environmental forcing of phytoplankton community structure and function in the Canadian High Arctic: contrasting oligotrophic and eutrophic regions. *Marine Ecology Progress Series*, *442*, 37-57.

Ardyna, M., & Arrigo, K. R. (2020). Phytoplankton dynamics in a changing Arctic Ocean. *Nature Climate Change*, *10*(10), 892-903.

Falardeau, M., Bennett, E. M., Else, B., Fisk, A., Mundy, C. J., Choy, E. S., ... & Moore, J. S. (2022). Biophysical indicators and Indigenous and Local Knowledge reveal climatic and ecological shifts with implications for Arctic Char fisheries. *Global Environmental Change*, *74*, 102469.

Florko, K. R., Tai, T. C., Cheung, W. W., Ferguson, S. H., Sumaila, U. R., Yurkowski, D. J., & Auger-Méthé, M. (2021). Predicting how climate change threatens the prey base of Arctic marine predators. *Ecology Letters*, *24*(12), 2563-2575.

Krumhardt, K. M., Lovenduski, N. S., Long, M. C., Luo, J. Y., Lindsay, K., Yeager, S., & Harrison, C. (2020). Potential predictability of net primary production in the ocean. *Global Biogeochemical Cycles*, *34*(6), e2020GB006531.

Lewis, K. M., & Arrigo, K. R. (2020). Ocean color algorithms for estimating chlorophyll a, CDOM absorption, and particle backscattering in the Arctic Ocean. *Journal of Geophysical Research: Oceans*, *125*(6), e2019JC015706.

Tai, T. C., Sumaila, U. R., & Cheung, W. W. (2021). Ocean acidification amplifies multi-stressor impacts on global marine invertebrate fisheries. *Frontiers in Marine Science*, *8*, 596644.

Tremblay, G., Belzile, C., Gosselin, M., Poulin, M., Roy, S., & Tremblay, J. É. (2009). Late summer phytoplankton distribution along a 3500 km transect in Canadian Arctic waters: strong numerical dominance by picoeukaryotes. *Aquatic Microbial Ecology*, *54*(1), 55-70.

Steinacher, M., Joos, F., Frölicher, T. L., Bopp, L., Cadule, P., Cocco, V., ... & Segschneider, J. (2010). Projected 21st century decrease in marine productivity: a multi-model analysis. *Biogeosciences*, *7*(3), 979-1005.

Steiner, N. S., Cheung, W. W., Cisneros-Montemayor, A. M., Drost, H., Hayashida, H., Hoover, C., ... & VanderZwaag, D. L. (2019). Impacts of the changing ocean-sea ice system on the key forage fish Arctic cod (Boreogadus saida) and subsistence fisheries in the western Canadian Arctic—Evaluating linked climate, ecosystem and economic (CEE) models. *Frontiers in Marine Science*, *6*, 179.

Usov, N., Radchenko, I., Smirnov, V., & Sukhotin, A. (2024). Joint seasonal dynamics of phytoplankton and zooplankton in the sub-Arctic White Sea. *Marine Ecology Progress Series*, *732*, 33-51.

---

## Author Comment (AC3)

**Response to Reviewer 3:**

**The authors present a numerical study of Arctic planktonic ecosystem productivity and trophic organization in response to major physical drivers, such as sea ice extent, light availability, temperature, several nutrient concentrations, stratification, advection, etc. Their study allows for a detailed exploration of a biogeochemical model of intermediate complexity that considers functional biodiversity and size-based dynamics as well, for a ~60-year time span at the scale of the whole Arctic domain. It leads to a rich harvest of numerical results that are well presented and analysed: despite the ambitious scope of the study, the paper does not feel too complex nor too long. I command the authors for having chosen an interesting balance between a more detailed inter-regional and seasonal approach, and approaches deployed at the larger interannual scale and trophic network level.**

**With all its qualities, I still think that there are some elements that should be addressed, especially regarding the comparison with observations (validation is too strong a word in the Arctic context).**

**First, regarding the comparison of simulated and observed surface chlorophyll a level (especially L270-274), I do not agree with the authors that "The comparison between model and satellite chlorophyll shows that in many cases the phenology of chlorophyll, if not absolute magnitudes, corresponded reasonably well". In *most* instances, the comparison is not convincing… the only instance where the *phenology* corresponds are the Nordic Sea (Fig. 3d), while the few instances where the *concentrations* match are again the Nordic Sea, Baffin Bay (Fig. 3a) and Central Arctic (Fig. 3e) for one month, while Chukchi and Barents Seas are relatively close for June-July, but with and obvious issue for the latter (increasing observed values in winter and fall…). I also stress that in all the chlorophyll a figures a log scale has been chosen, which obviously tends to downplay the discrepancies.**

**It does not mean at all that the modelling study is not valid, since the model behaves logically and that satellite-based observations of Arctic chlorophyll a are notoriously challenging to handle. I think though that the authors should not downplay this difficult comparison; I'd rather prefer them to recognize it and try to refer to other studies that might have run into the same issue (e.g. Popova et al. 2012. 10.1029/2011JC007112; my reference here is old, the authors might know more recent ones). Moreover, I think the authors could provide an estimation, at least in the discussion, of how the use of a different color algorithm that takes into account the high CDOM content of these waters could change the satellite-based biomass estimates (e.g. Li et al. 2023 10.1364/OE.500340).**

- Thank you for your comment, which is similar to Reviewer 1 and 2's concerns about the comparison with satellite chlorophyll estimates. In the revised paper, we will use chlorophyll estimated with an algorithm tailored to the Arctic Ocean (Lewis and Arrigo, 2020; https://doi.org/10.1029/2019JC015706). We updated Figures 2 and 3 and updated a

detailed discussion of model-satellite chlorophyll comparison with this satellite product, as detailed in our response to Reviewer 1. Note, that due to different satellite grids, we have modified the seasonal comparison following regional masks by Lewis & Arrigo 2020 shown in Figure 1.

[Figure]

Figure 1. Map of the Arctic Ocean divided into ten sections included in the analysis, following Lewis & Arrigo (2020) regional mask. Additionally, symbols represent three grid cells selected throughout the Arctic Ocean for growth limitation analysis. The cross symbol shows the Western Nordic Seas ($68.5^o$N, $348^o$E), the star symbol shows the central Arctic Ocean location ($85.5^o$N, $200^o$E), and the triangle symbol shows the Chukchi Sea location ($68.5^o$N, $168^o$W).

**Second, starting at section 3.3 approx., it seems to me that about half of the results shown are not / cannot be directly compared to observations, from the slopes of the size-structured community to the trophic position of the zooplankton groups. It would strengthen the paper if the authors could find information on the slope & intercept of the size structure of plankton community, from Tara Ocean, Mozaik expedition or other sources like this, if possible. What is certainly possible, though, is for the authors to present some ideas of experiments that should be done to test the many hypotheses their model has generated! I think it would be very useful for the community of Arctic oceanographers as a whole, since there are so many outcomes of their model that could provide guidance for future in situ experiments.**

- Thank you for your comment. We agree that comparing the size spectra relationships with observations would be desirable. There is a new database on plankton size structure with global coverage, prepared by Dugenne et al. (2024) (https://essd.copernicus.org/articles/16/2971/2024/) . The Arctic data in the Pelagic Size Structure database (PSSdb) is from the Tara Polar Circle expedition. However, the

availability of data for the Arctic Ocean is limited, with substantial data only available for 2009, 2011, 2013 and 2019. Below we show the slope of the normalized biomass size spectra (NBSS) from three different instruments, the Imaging FlowCytoBot (IFCB; primarily phytoplankton and surface only data), Scanners (e.g., ZooScan, primarily zooplankton size ranges and 0-200m depth-integrated), and Underwater Vision Profiler (UVP; primarily zooplankton size ranges and 0-200m depth-integrated). Compared to the size abundance relationship used in our study, the NBSS is an alternate way of computing a size spectra, with similar implications (lower numbers = steeper slopes, with more small plankton compared to large plankton, and vice versa).

- However, because of the very limited data availability in the Arctic and the fact that the PSSdb data only overlaps with our model simulations in the year 2009, with the UVP instrument, we do not feel that a formal comparison would be appropriate at this time, but hope that such a comparison could be done in a future study where the dates of the model simulation and the observations have a greater overlap.

[Figure]

- Research into zooplankton trophic positions, such as the study by Choi et al. (2020) on trophic dynamics before and after polar night, reveals seasonal changes in diet and survival strategies in polar regions influenced by varying solar radiation and oceanographic conditions. Nitrogen isotope analysis in their study indicated stable trophic positions for species like *Parasagitta elegans (chaetognath)* and *Calanus spp. (copepod)*, whereas others like *Oithona similis* (copepod) showed elevated trophic positions post-polar night, potentially due to sustained energy intake enabled by adaptable feeding behaviors (Berge et al., 2015). Given the food-limited environment during polar night, most zooplankton are likely to reduce their food consumption (Grigor et al., 2014) or enter a period of rest until conditions improve. In contrast, opportunistic zooplankton can easily shift from algae to

protists (Iversen and Seuthe, 2011). Therefore, the observed seasonal variation in trophic positions in our study aligns with a higher representation of opportunistic zooplankton in our model simulations.

- We incorporated some of this text in the manuscript in L364-366: "..., and reflect seasonal variations in diet and survival strategies of more opportunistic zooplankton with adaptable feeding behaviors (Berge et al. 2015), as observed by Choi et al. (2020) in *Oithona similis* (copepod) exhibiting elevated trophic positions post-polar night."

**Detailed comments**

1. **L76-80: a simple schematic of the different types and groups of plankton would help.**
   - Figure 1 in Negrete-Garcia et al. 2022, the paper that first documented the model, has a nice figure describing the model structure. We will add text referring readers to that paper.
   - L78: "(Negrete-García et al., 2022, Figure 1)" .

2. **L99: "[…] is suited to study Arctic Ocean dynamics." I think a few more details on issues specific to the Arctic ecosystem, such as usual temperature, light and nitrate limitation should be provided before stating that.**
   - Thank you for your comment. We added the following text (L99-102):"This balance is crucial for studying the Arctic Ocean, where complex interactions between plankton communities and physical environmental factors play significant roles in ecosystem dynamics. The model's ability to represent diverse plankton functional types and their responses to varying nutrient and light conditions enable a nuanced understanding of Arctic biogeochemical cycles and food web structures."

3. **L140: while I do not suggest the authors modify the MARBL-SPECTRA model, I would like them to recognize that the choice of a type II functional response is not always optimal and can have a destabilizing influence in NPZD-type models (e.g. Gentleman & Neuheimer 2008, 10.1093/plankt/fbn078; Flynn & Mitra 2016, 10.3389/fmars.2016.00165).**
   - In the development process for MARBL-SPECTRA (described in Negrete-Garcia et al. 2022), we conducted extensive testing, tuning, and sensitivity analyses on the grazing formulation, including testing alternative functional forms (e.g., Holling Type III functional response). In total, we conducted nearly 400 test simulations, some of which is shown in the formal sensitivity analysis that we conducted for Negrete-Garcia et al. 2022 (see supplemental information in that manuscript). In our simulations and testing, we found no evidence that the grazing functional response induced oscillations and instabilities. Further, we found that the Holling Type II functional form performed the best in simulating the global biogeography of a number of key metrics, including chlorophyll, phytoplankton size structure, macronutrients, and zooplankton biomass.
   - Nonetheless, we acknowledge the reviewer's point and recognize that the grazing response choice may not reflect reality. Thus, we added the following text in the

model limitations section (L501-507): "Additionally, there are significant uncertainties and omissions in the representation of zooplankton, their grazing, and population dynamics in MARBL-SPECTRA. Zooplankton grazing (including food web structure, grazing parameters, and functional responses) is one of the largest sources of uncertainty in ocean biogeochemical models (Rohr et al. 2023, Hansen et al. 1997, Gentleman and Neuheimer 2008), and in this model, we do not resolve key zooplankton life histories or migration dynamics such as dormancy or diapause, which are important traits that allow organisms to survive unfavorable environmental conditions in the Arctic (Baumgartner and Tarrant, 2017)."

4. **L189: why integrating over 150m?**
   ○ In CESM, the default model diagnostics for most marine ecosystem variables are provided for the top 150 m. This is a practical choice, as it balances the need for increased diagnostics with the need to save disk space for the diagnostic-heavy ocean biogeochemical model. Thus, some of the key model diagnostics are depth-integrated biological variables over the top 150 meters, which typically encompass the euphotic zone. We chose to keep this default for a number of reasons: integrating over this depth ensures that the model captures the most significant biological activity related to carbon fixation and nutrient cycling. Additionally, the top 150 meters often include the mixed layer, where physical mixing distributes heat, nutrients, and gasses, further influencing biological and chemical processes. By focusing on this depth range, the model can more accurately represent the complex interactions and gradients that drive ocean productivity and biogeochemical cycles, while making sure we stay within our disk space quota.

5. **L251-253: I am not sure whether the authors speak about a usual spatial trend or a temporal trend resulting from the impacts of climate change?**
   ○ Thank you for your comment. We realize that the text was not very clear in distinguishing between spatial and temporal trends. In this context, we are referring to a temporal trend throughout the hindcast simulation. To clarify this, we added a reference to the Chukchi Sea anomaly plots, which illustrate these temporal changes more clearly (L256-257): "Specifically, the model illustrated a trend towards a more oligotrophic western Arctic Ocean basin (Fig. 8e)."

6. **L295: where do the nutrients come from? Remineralization or advection? Both?**
   ○ In this context, we mean that nutrients accumulate over the winter due to remineralization and lower consumption by microbes. However, horizontal and vertical transport also play a role.

7. **L451: this first sentence seems to contradict what was just said in the previous paragraph, probably because this model does not take into account sympagic production and its export towards the benthos in spring.**
   ○ Thank you for your comment. We acknowledge that the text may appear contradictory at first glance. In the previous paragraph, we discussed the general

trends observed over the hindcast period, highlighting the variability in fisheries production across different regions of the Arctic Ocean and the role of mesozooplankton biomass and carbon export to the benthos.

The sentence in L451 refers to specific years with lower ice fraction and higher temperatures supporting higher fish production. This observation is consistent with the overall trend that reduced ice cover and increased temperatures can enhance primary productivity and, consequently, fish production in some regions.

However, as you rightly pointed out, our model does not explicitly account for sympagic production and its export towards the benthos in spring. The absence of this factor in our model may lead to an underestimation of the contributions of ice-associated production to benthic and pelagic ecosystems during periods of high sympagic activity. This limitation might explain the perceived contradiction, as the model primarily captures pelagic processes and their direct impacts on fisheries production.

To clarify this, we incorporated the following text in the Model study limitations section (L497-L500): "Sea ice algae play crucial roles in shaping sea-ice associated ecosystems and biogeochemistry (Kohlbach et al., 2016), and their absence in our model may lead to an underestimation of the contributions of ice-associated production to benthic and pelagic ecosystems during periods of high sympagic activity."

8. **L465-466: please provide in one or two sentences indications on how this could have affected your conclusions, much like you did in the following paragraph.**
   ○ Thank you for your comment. We included the following in lines L507-509 "The omission of these traits may lead to an incomplete representation of zooplankton population dynamics and their seasonal availability as predators and prey. Consequently, this limitation influenced the underestimation of mesozooplankton biomass (Fig. 2o), potentially influencing our assessments of trophic interactions and fisheries production."

**References:**

Baumgartner, M. F., & Tarrant, A. M. (2017). The physiology and ecology of diapause in marine copepods. *Annual review of marine science*, *9*(1), 387-411.

Berge, J., Renaud, P. E., Darnis, G., Cottier, F., Last, K., Gabrielsen, T. M., ... & Falk-Petersen, S. (2015). In the dark: a review of ecosystem processes during the Arctic polar night. *Progress in Oceanography*, *139*, 258-271.

Choi, H., Ha, S. Y., Lee, S., Kim, J. H., & Shin, K. H. (2020). Trophic dynamics of zooplankton before and after polar night in the Kongsfjorden (Svalbard): Evidence of trophic position estimated by δ15N analysis of amino acids. *Frontiers in Marine Science*, *7*, 489.

Dugenne, M., Corrales-Ugalde, M., Luo, J. Y., Kiko, R., O'Brien, T. D., Irisson, J. O., ... & Vilain, M. (2024). First release of the Pelagic Size Structure database: global datasets of marine size spectra obtained from plankton imaging devices. *Earth System Science Data*, *16*(6), 2971-2999.

Gentleman, W. C., & Neuheimer, A. B. (2008). Functional responses and ecosystem dynamics: how clearance rates explain the influence of satiation, food-limitation and acclimation. *Journal of Plankton Research*, *30*(11), 1215-1231.

Grigor, J. J., Søreide, J. E., & Varpe, Ø. (2014). Seasonal ecology and life-history strategy of the high-latitude predatory zooplankter Parasagitta elegans. *Marine Ecology Progress Series*, *499*, 77-88.

Hansen, P. J., Bjørnsen, P. K., & Hansen, B. W. (1997). Zooplankton grazing and growth: Scaling within the 2-2,-μm body size range. *Limnology and oceanography*, *42*(4), 687-704.

Kohlbach, D., Schaafsma, F. L., Graeve, M., Lebreton, B., Lange, B. A., David, C., ... & Flores, H. (2017). Strong linkage of polar cod (Boreogadus saida) to sea ice algae-produced carbon: evidence from stomach content, fatty acid and stable isotope analyses. *Progress in Oceanography*, *152*, 62-74.

Lewis, K. M., & Arrigo, K. R. (2020). Ocean color algorithms for estimating chlorophyll a, CDOM absorption, and particle backscattering in the Arctic Ocean. *Journal of Geophysical Research: Oceans*, *125*(6), e2019JC015706.

Li, J., Matsuoka, A., Pang, X., Massicotte, P., & Babin, M. (2024). Performance of Algorithms for Retrieving Chlorophyll a Concentrations in the Arctic Ocean: Impact on Primary Production Estimates. *Remote Sensing*, *16*(5), 892.

Negrete-García, G., Luo, J. Y., Long, M. C., Lindsay, K., Levy, M., & Barton, A. D. (2022). Plankton energy flows using a global size-structured and trait-based model. *Progress in Oceanography*, *209*, 102898.

Rokkan Iversen, K., & Seuthe, L. (2011). Seasonal microbial processes in a high-latitude fjord (Kongsfjorden, Svalbard): I. Heterotrophic bacteria, picoplankton and nanoflagellates. *Polar biology*, *34*, 731-749.

Rohr, T., Richardson, A. J., Lenton, A., Chamberlain, M. A., & Shadwick, E. H. (2023). Zooplankton grazing is the largest source of uncertainty for marine carbon cycling in CMIP6 models. *Communications Earth & Environment*, *4*(1), 212.